# Different effects of anthropogenic emissions and aging processes on the mixing state of soot particles in the nucleation and accumulation modes

Yuying Wang[1,2], Rong Hu[1,2], Qiuyan Wang[1], Zhanqing Li[3], Maureen Cribb[3], Yele Sun[4], Xiaorui Song[1], Yi Shang[1], Yixuan Wu[1], Xin Huang[1], Yuxiang Wang[1]

[1] Key Laboratory for Aerosol-Cloud-Precipitation of China Meteorological Administration, School of Atmospheric Physics, Nanjing University of Information Science & Technology, Nanjing 210044, China
[2] State Key Laboratory of Remote Sensing Science, College of Global Change and Earth System Science, Beijing Normal University, Beijing 100875, China
[3] Earth System Science Interdisciplinary Center, Department of Atmospheric and Oceanic Science, University of Maryland, College Park, MD, USA
[4] State Key Laboratory of Atmospheric Boundary Layer Physics and Atmospheric Chemistry, Institute of Atmospheric Physics, Chinese Academy of Sciences, Beijing, 100029, China

Correspondence to: Yuying Wang (yuyingwang@nuist.edu.cn)

**Abstract**. In this study, the mixing state of size-resolved soot particles and their influencing factors were investigated based on a five-month aerosol volatility measurement at a suburban site (Xingtai, XT) in the central North China Plain (NCP). The volatility and mixing state of soot-containing particles at XT were complex caused by multiple pollution sources and various aging processes. The results suggest that anthropogenic emissions can weaken the mean volatility of soot-containing particles and enhance their degree of external mixing. There were fewer externally mixed soot particles in warm months (June, July, and August) than in cold months (May, September, and October). Monthly variations in the mean coating depth ($D_{c,mean}$) of volatile matter on soot particles showed that the coating effect was stronger in warm months than in cold months, even though aerosol pollution was heavier in cold months. Moreover, the volatility was stronger, and the degree of internal mixing was higher in nucleation-mode soot-containing particles than in accumulation-mode soot-containing particles. Relationships between $D_{c,mean}$ and possible influencing factors [temperature ($T$), relative humidity (RH), and particulate matter with diameters ranging from 10 to 400 nm] further suggest that high ambient $T$ and RH in a polluted environment could promote the coating growth of accumulation-mode soot particles. However, high ambient $T$ but low RH in a clean environment were beneficial to the coating growth of nucleation-mode soot particles. Our results highlight the diverse impact of anthropogenic emissions and aging processes on the mixing state of soot particles in different modes, which should be considered separately in models to improve the simulation accuracy of aerosol absorption.

## 1. Introduction

Aerosols are mixed liquid and solid particles suspended in the atmosphere. Some aerosols are directly produced from natural or anthropogenic sources (i.e., primary aerosols), and the rest are indirectly transformed from gas precursors through atmospheric chemical reactions (i.e., secondary aerosols). The newly formed particles can grow or shrink through various aging processes (e.g., condensation, coagulation, volatilization, chemical reactions). Aerosol physicochemical properties (number concentration, shape, mixing state, optical properties, among others) are thus highly variable. This is one of the reasons why aerosols are highly uncertain in climate change assessments (Bond et al., 2013; Seinfeld et al., 2016; Bellouin et al., 2020; Christensen et al., 2021). Although great efforts have been made to understand aerosol optical properties, the uncertainty of radiative forcing caused by aerosols is still two to three times that of greenhouse gases (IPCC, 2021).

Aerosols can affect the earth-atmosphere radiation balance by scattering or absorbing shortwave and longwave radiation, which is called the aerosol direct climate effect or aerosol-radiation interactions. Many factors, such as aerosol chemical composition, mixing state, and ambient relative humidity (RH), have complex impacts on aerosol-radiation interactions (e.g., Twohy et al., 2009;

Kuniyal and Guleria, 2019; Ren et al., 2021). According to the sixth IPCC report, the total direct radiative forcing caused by anthropogenic aerosols is generally negative. However, light-absorbing carbonaceous particles (LAC) have a warming effect on climate (Ramana et al., 2010; Gustafsson and Ramanathan, 2016), which can partly offset the cooling effect caused by scattering aerosols, such as sulfate. Black carbon (BC) is the most important LAC compound. Some experiments have suggested that BC in urban polluted environments can play an important role in pollution formation and development. The internal mixing of BC with secondly formed matter could also greatly enhance light absorption (Peng et al., 2016; Zhou et al., 2017). Moreover, BC is mostly emitted as soot from anthropogenic sources (incomplete fossil fuel combustion and biomass burning) (Novakov et al., 2003). Soot particles are abundant in both nucleation and accumulation modes (Li et al., 2011; Levy et al., 2013; La Rocca et al., 2015; Hu et al., 2021; Zhang et al., 2021).

The online measurement instruments quantifying the mixing state of BC-containing particles are limited. Based on the measurement of single-particle soot photometer (SP2), Wu et al. (2017) indicated that the mass of refractory black carbon ($r$BC) had an approximately lognormal distribution as a function of the volume-equivalent diameter (VED) in Beijing. Yu et al. (2020) suggested that the mixing state of $r$BC particles was related to air pollution levels and air mass sources. Zhang et al. (2021) further indicated that meteorological conditions had a large impact on the mixing state of $r$BC particles. Moreover, the Aerodyne soot particle aerosol mass spectrometer (SP-AMS) can also be used to study the mixing state of $r$BC. For example, J. Wang et al. (2019) found that the formation of secondary aerosols through photochemical and aqueous chemical reactions was responsible to the coating of $r$BC based on the measurement of SP-AMS in winter Beijing. However, the lower observation limit of particle size by SP2 and SP-AMS is larger than ~70 nm. Therefore, they cannot quantify the mixing state of BC-containing particles in the small nucleation mode.

Aerosol volatility refers to the shrinking extent of particles at a certain temperature. The mixing state of soot particles or tarballs is closely related to aerosol volatility at high temperatures (Philippin et al., 2004; Wehner et al., 2009; Adachi et al., 2018, 2019). Most primary soot particles from anthropogenic sources are refractory, hydrophobic, and externally mixed. In a polluted environment, primary soot particles are easily transformed to internally mixed particles through certain coating processes in the atmosphere (Cheng et al., 2012; Peng et al., 2016; F. Zhang et al., 2020). However, coating matter is generally non-refractory because most of the matter consists of secondary chemical species, such as organics, sulfate, and nitrate (Philippin et al., 2004; Hong et al., 2017). This is why aerosol volatility can characterize the mixing state of soot particles in polluted environments (Wehner et al., 2009; Hossain et al., 2012; S. Zhang et al., 2016). A volatility tandem differential mobility analyzer (VTDMA) is usually used to quantify aerosol volatility by measuring the change

in particle size at a set temperature. Aerosol volatility measured by a VTDMA at a high temperature
(> 280°C) can be used to study the mixing state of soot particles (Philippin et al., 2004; Wehner et
al., 2009; Y. Zhang et al., 2016; Wang et al., 2017). Meanwhile, VTDMA measurements are based
on the aerosol number concentration, which is always high in the nucleation mode in the ambient.
Therefore, VTDMA can quantify the mixing state of nucleation-mode soot particles.

Over the past years, several studies have reported the volatility and mixing state of particles based

on VTDMA measurements in the North China Plain (NCP). For example, Wehner et al. (2009)
found that the mixing state of soot particles in Beijing and its surrounding region varied, especially
between new particle formation days and heavily polluted days. Using the same VTDMA and
aerosol optical data, Cheng et al. (2009) conducted an aerosol optical closure study, finding that soot
aging was rapid at the Yufa site south of Beijing. The coating on soot particles can enhance aerosol
absorption and scattering coefficients by a factor of 8 to 10 within several hours due to secondary
processing during the daytime, which is the combined effect of the increased thickness of the coating
shell and the transition of soot from an externally mixed state to a coated state. Cheng et al. (2012)
further indicated that aging and emissions were two competing factors in the mixing state of soot
particles. Based on VTDMA measurement data collected in 2015, Wang et al. (2017) indicated that
strict emission control measures implemented in Beijing and surrounding areas could enhance the
volatility of soot-containing particles and their degrees of external mixing. At another regional site
(Xianghe) in the northern part of the NCP, S. Zhang et al. (2016) found that the mixing state of
ambient particles was complex with different volatilities. Furthermore, Y. Zhang et al. (2016)
suggested that the average shell-to-core ratio and absorption enhancement ($E_{ab}$) of ambient BC was
2.1–2.7 and 1.6–1.9, respectively.

These studies imply that anthropogenic emissions play an important role in the volatility and

mixing sate of soot-containing particles and that the coating on soot particles can greatly enhance
aerosol absorption. However, these studies were based on data collected during short-term
observational periods in the northern part of the NCP and they did not distinguish the factors
influencing the mixing state of nucleation- and accumulation-mode soot particles. Recent studies
(Y. Wang et al., 2018, 2019, 2021) have shown that anthropogenic sources and aerosol aging
processes are various in the north and central-south NCP, leading to diverse aerosol physiochemical
properties between these regions in different seasons. More research about the mixing state of soot
particles in the central-south NCP is needed to improve the accuracy of modeled aerosol optical
properties.

This study investigates for the first time the volatility and mixing state of nucleation- and

accumulation-mode soot-containing particles in the warm and cold seasons based on one
comprehensive field campaign that took place in the central NCP, lasting five months. Exploring

factors influencing the volatility and mixing state of soot-containing particles in this study will improve the accuracy of modeled aerosol optical properties in the central NCP. This paper is organized as follows. Section 2 introduces the sampling site, instruments, and data analysis. Section 3 presents the results and discussion, including meteorological conditions, aerosol pollution levels, changes in volatility and mixing state of soot-containing particles, and their influencing factors. Section 4 gives conclusions and summarizes the study.

## 2.    Sampling site, instruments, and data analysis

### 2.1    Sampling site

Data used in this study were collected at the National Meteorological Basic Station ($37°11'$N, $114°22'$E, 180 m above sea level) in Xingtai (XT), China, equipped with a variety of meteorological observation instruments. The measured meteorological variables including ambient temperature, relative humidity (RH), wind direction and speed were used in this study. Y. Wang et al. (2018) reported that this site was located in a polluted area of the central-south NCP, influenced by multiple anthropogenic sources, such as industrial coal firing, fossil-fuel burning, agricultural activities, and household emissions. The long-distance transport of pollutants also influences the air quality at XT. Previous studies have suggested that air pollution at XT represents well regional pollution characteristics in the central NCP, east of the Taihang Mountains (Y. Zhang et al., 2018; Y. Wang et al., 2018). A comprehensive field campaign named the Atmosphere-Aerosol-Boundary layer-Cloud ($A^2BC$) Interaction Joint Experiment was carried out at XT from May to October of 2016. Y. Wang et al. (2018) and Li et al. (2019) provide details about the XT site and the $A^2BC$ campaign. Here, over five months of aerosol observational data, including particle number size distribution (PNSD), aerosol volatility, and BC mass concentration, were used to analyze the volatility and mixing state of soot particles and their influencing factors.

### 2.2    Instruments

#### 2.2.1    Measuring PNSD and aerosol volatility

The tandem differential mobility analyzer (TDMA) system is widely used to measure the change in particle size under special conditions, e.g., high humidity, high temperature, and chamber chemical reactions (Swietlicki et al., 2008). In this campaign, the VTDMA system was used to measure aerosol volatility at 300$°$C. The inlet air sample was first dried by a Nafion$^{TM}$ dryer to low RH (< 30%), then neutralized by a soft X-ray neutralizer (model 3088, TSI Inc.; Fig. 1a). Afterwards, quasi-monodisperse aerosols (Fig. 1b) with a certain dried diameter ($D_d$) were split by the first differential mobility analyzer (DMA1). In this campaign, $D_d$ was set to 40, 80, 110, 150, 200, and 300 nm. An automated valve located after the DMA1 had two outlet lines. Line 1 directly accessed

the water-based condensation particle counter (WCPC, model 3787, TSI Inc.), measuring the
number concentration of particles ranging from 10 to 400 nm. Line 2 accessed a heating tube,
vaporizing volatile materials at a controlled high temperature (300°C in this study). The ratio of
particle size after volatilization [$D_p(T)$] to $D_d$ is defined as the aerosol shrink factor (i.e., $SF = D_p(T)$
/ $D_d$). After heating, residual aerosols were generally polydisperse nonvolatile particles (Fig. 1c).
The second DMA (DMA2) and WCPC were used to measure the number size distribution of
nonvolatile particles, measuring the distribution function of $SF$ ($SF$-MDF). Finally, the probability
density function of $SF$ ($SF$-PDF) was retrieved using the TDMAfit algorithm (Stolzenburg and
McMurry, 1988; Stolzenburg and McMurry, 2008).
In this study, we assume that the shape of all particles follows the core-shell model (nonvolatile
core and volatile shell; Fig. 1b). Residual particles after volatilization have different-sized
nonvolatile cores (Fig. 1c). Previous studies have suggested that residual particles at 300°C mainly
consist of soot (Philippin et al., 2004; Wehner et al., 2009). Aerosol volatility measured by the
VTDMA in this study can thus reflect the degree of mixing state of soot particles.

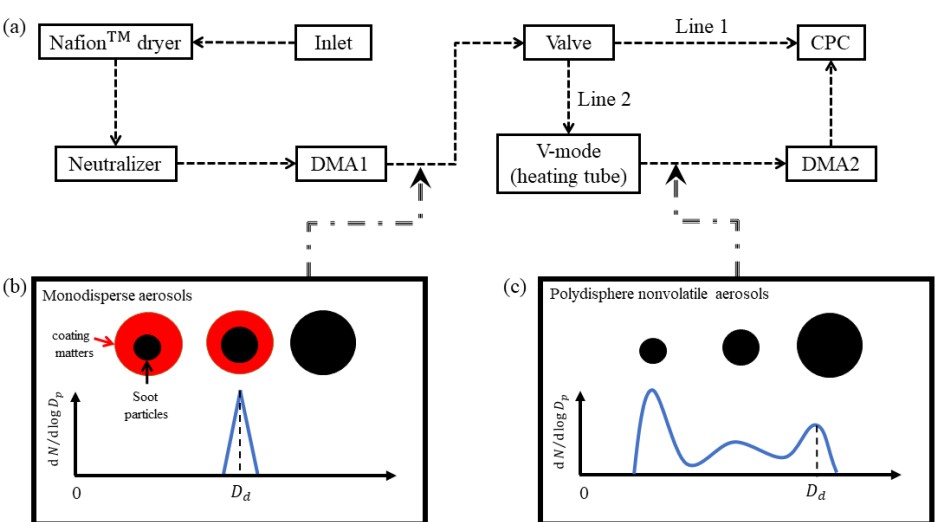


**Figure 1**. Schematic diagram of the volatility tandem differential mobility analyzer used in this
study.

2.2.2    Measuring BC
In this campaign, a seven-wavelength aethalometer (model AE-33, Magee Scientific Corp.) was
used to measure the mass concentration of BC ($M_{BC}$). After calibration, the sampling flow rate of
the AE-33 was 5.0 L min$^{-1}$. A cyclone with particulate matter (diameters = 2.5 μm, or PM$_{2.5}$) was
used in the sample inlet. Aerosol particles were collected on filter tape through a spot, and the
instantaneous concentration of optically absorbing aerosols was retrieved from the rate of change
of the attenuation of light transmitted through the filter. The wavelength channels of the AE-33 were

370, 470, 525, 590, 660, 880, and 940 nm. According to the manufacturer's instructions, the $M_{BC}$ is calculated from the change in optical attenuation at channel 6 (i.e., 880 nm) in the selected time interval using the mass absorption cross section (MAC) of 7.77 m$^2$ g$^{-1}$. The dependency of MAC on BC coating may introduce some uncertain in calculating MAC (Drinovec et al., 2015).

2.2.3   VTDMA data analysis

The retrieved $SF$-PDF ($c(D_d, SF)$) is normalized as $\int c(D_d, SF)\,dSF = 1$. The ensemble mean shrink factor ($SF_{mean}$) is then calculated as

$$SF_{mean}(D_d) = \int_0^\infty SF \cdot c(D_d, SF)\,dSF \ . \qquad (1)$$

Particles can be classified into several volatile groups according to different $SF$ ranges (Y. Wang et al., 2017). The number fraction ($NF$) for each volatile group with the $SF$ boundary of [a, b] is calculated as

$$NF(D_d) = \int_a^b c(D_d, SF)\,dSF \ . \qquad (2)$$

Based on the core-shell assumption, the coating depth ($D_c$) of soot particles is defined as the depth of shell materials (i.e., shell depth). According to the definition of $SF$, $D_c$ for the particle ($D_d, SF$) can be calculated as

$$D_c(D_d, SF) = \frac{D_d}{2}(1 - SF). \qquad (3)$$

The ensemble mean $D_c$ ($D_{c,mean}$) using the normalized $SF$-PDF data is then calculated as

$$D_{c,mean}(D_d) = \int_0^\infty D_c(D_d, SF) \cdot c(D_d, SF)\,dSF. \qquad (4)$$

## 3.   Results and discussion

### 3.1   Meteorological conditions and aerosol pollution levels

Figure 1a-b shows the time series of ambient temperature ($T$), RH, and wind direction and speed (WD and WS, respectively) during the campaign. Monthly changes in $T$ are clearly seen (Fig. 2a). Average $T$s in warm (June, July, and August) and cold (May, September, and October) months were 25.73±3.80 and 19.0±5.74$^o$C, respectively. The meteorological variables changed periodically in cold months but not in warm months, which is caused by the cold fronts in cold months. Figure 2a also suggests that RH was higher in July and August than in other months.

Figure 2b shows that the wind changed significantly in different months at XT. Monthly wind rose diagrams (Fig. S1) indicate that northwest winds prevailed in all months, caused by the special terrain around XT (Y. Zhang et al., 2018). In July, weak southeast winds were also present, beneficial to the accumulation of air pollutants due to the stable atmospheric environment. In August,

the other prevailing wind was from the north, which was beneficial for atmospheric diffusion.

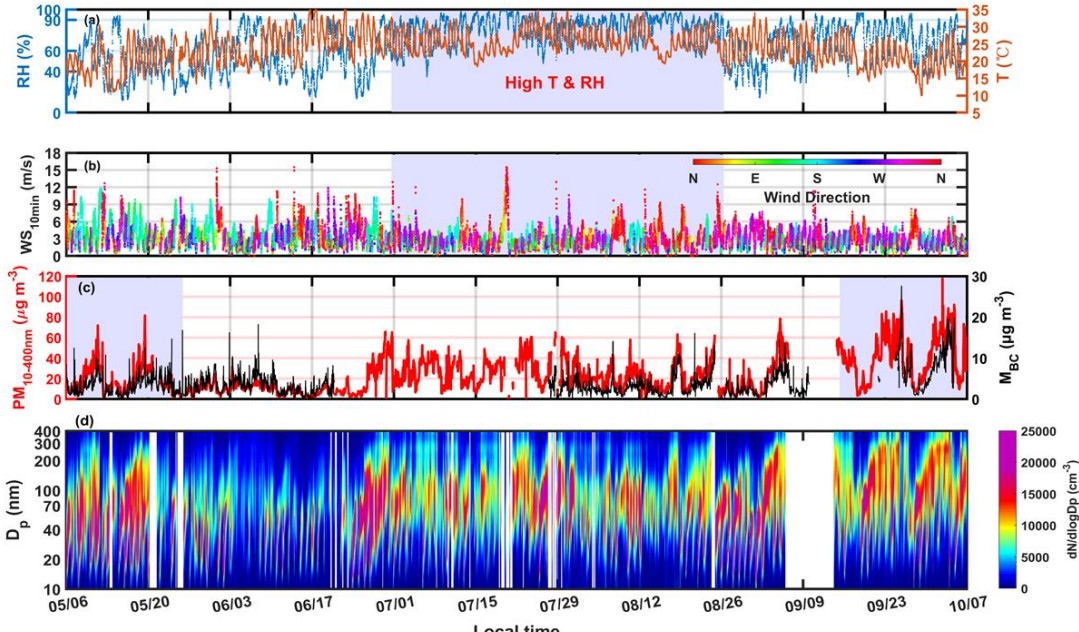

**Figure 2**. Time series of (**a**) ambient relative humidity (RH; unit: %) and temperature ($T$; unit: ºC),
(**b**) wind direction (WD) and 10-minute-averaged wind speed (WS; unit: m s$^{-1}$), (**c**) mass
concentration of 10–400 nm particles (PM$_{10\text{-}400}$, in red; unit: μg m$^{-3}$), assuming that the aerosol
density is 1.6 g cm$^{-3}$, and mass concentration of black carbon ($M_{BC}$, in black; unit: μg m$^{-3}$), and (**d**)
particle number size distribution at the Xingtai site from 6 May 2016 to 6 October 2016.

In this study, the total mass concentration of 10–400-nm particles (PM$_{10\text{-}400}$) (Fig. 2c) was
calculated using PNSD data (Fig. 2d), assuming that the aerosol density was 1.6 g cm$^{-3}$ (Y. Wang
et al., 2017). The average PM$_{10\text{-}400}$ concentrations in warm and cold months were 19.68±13.58 and
29.79±21.37 μg m$^{-3}$, respectively, indicating much higher aerosol pollution in cold months than in
warm months. In cold months, PM$_{10\text{-}400}$ accumulated periodically as accumulation-mode ($D_p >$ 100
nm) particles increased. This is closely related to cyclic changes in general atmospheric circulation,
reflected by the cycle of winds (Fig. 2b). However, PM$_{10\text{-}400}$ was lower in May than in September
and October, likely due to the weaker particle growth in May. During warm months, PM$_{10\text{-}400}$
reached its lowest value in June with the lowest number concentration of accumulation-mode
particles of all months (Fig. S2), suggesting that meteorological conditions in June were not
conducive to particle growth. The high $T$ and RH in July and August were beneficial to particle
growth by promoting atmospheric photochemical and liquid chemical reactions (Z. Wu et al., 2018;
Peng et al., 2021). Figure 2c suggests that PM$_{10\text{-}400}$ was much higher in July and August than in June,
although the mass concentrations of black carbon ($M_{BCS}$) in these months were considerable.
However, PM$_{10\text{-}400}$ was lower in August than in July, likely because of the better atmospheric
diffusion conditions (more and stronger northerly winds) in August. Figure 2c also shows that
changes in $M_{BC}$ and $PM_{10-400}$ were similar, suggesting the possible role of BC in the formation
processes of aerosol pollution. Recently, F. Zhang et al. (2020) demonstrated that BC-catalyzed
sulfate formation involving $NO_2$ and $NH_3$ plays an important role in the formation of haze events.

3.2     Monthly and diurnal variations in *SF*-PDF
Figure 3 shows the size-resolved mean *SF*-PDFs at XT. In general, *SF*-PDFs had three peak
modes, namely, at $SF \approx 0.4$ [very volatile (VV) mode], 0.6 [slightly volatile (SV) mode], and 0.9
[nonvolatile (NV) mode]. The trimodal distributions of *SF*-PDFs at XT in the central NCP differ
from those at sites in the northern NCP (S. Zhang et al., 2016; Y. Wang et al., 2017), implying
highly complex volatility and mixing state of soot particles at XT. Note that the *SF*-PDF of 40-nm
particles has a quasi-unimodal distribution pattern peaked at VV mode, with low fractions of NV-
and SV-mode particles. This means that the residual soot size of most 40-nm particles after heating
at 300ºC was about 16 nm. These tiny soot particles are mainly from anthropogenic sources such
as vehicle emissions (La Rocca et al., 2015; Hu et al., 2021). Extremely low-volatile organics are
another possible component in this size. However, these extremely low-volatile organics are
mainly formed in the forest area (Ehn et al., 2014). XT is located in the severe polluted area with
kinds of anthropogenic sources. It is excepted that there are many nucleation-mode soot particles
in the ambient in this region. In addition, previous studies have indicated that most NV-mode
particles are externally mixed soot particles (Cheng et al., 2012; Cheung et al., 2016). This
suggests that soot-containing particles in nucleation mode (represented by 40-nm particles) in this
study had strong volatility and a high degree of internal mixing.
Figure 3 also suggests that the fraction of NV-mode particles increased with increasing
particle size, indicating a higher fraction of externally mixed soot particles in accumulation mode.
This is related to the primary size of soot particles. Some studies suggest that most freshly emitted
refractory particles (like BC) are primarily in accumulation mode. For example, Levy et al. (2013)
reported that fresh BC was mostly in the 150–240 nm size range, while Wu et al. (2017) reported
that refractory BC size distribution measurements in Beijing peaked at about 200 nm, with a
secondary less significant mode at about 600 nm.

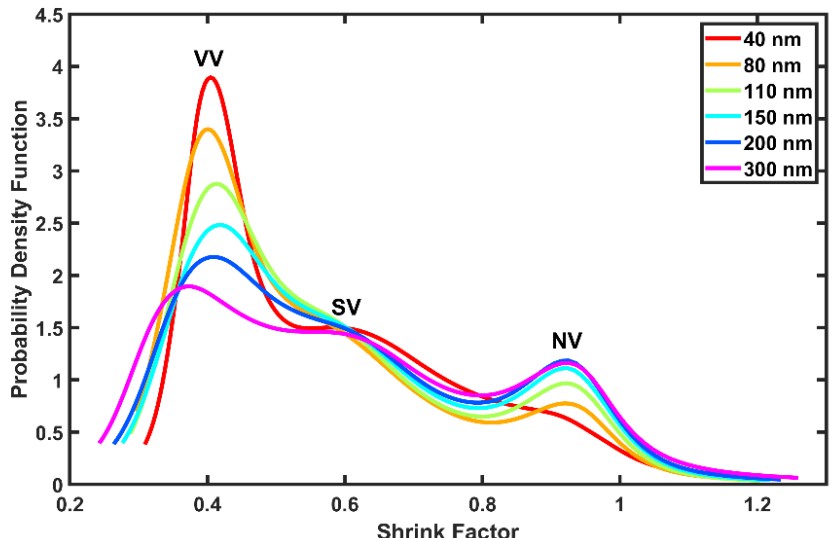

**Figure 3.** Size-resolved mean probability density functions of the shrink factor at different wavelenghts. VV stands for "very volatile", SV stands for "slightly volatile", and NV stands for "non-volatile".

Figure 4a-b shows that VV-mode fractions in the *SF*-PDFs of 40-nm and 80-nm particles were higher in warm months than in cold months, indicating that nucleation-mode soot particles were more volatile in warm months. Our previous study has shown that new particle formation (NPF) events occurred frequently at XT (Y. Wang et al., 2018). Wehner et al. (2009) reported that most newly formed matter is composed of organics and sulfate, easily volatized at 300°C. Li et al. (2011) indicated that the tiny soot particles embedded in sulfates could promote particle growth during NPF events in the NCP. All this implies that coating by newly formed secondary matter was the possible reason for the high volatility of nucleation-mode soot-containing particles in warm months. For accumulation-mode (110–300 nm) particles (Fig. 4c-f), monthly changes in *SF*-PDF patterns are clearly seen. In general, *SF* peak values of the VV mode were smaller (meaning a thicker coating of volatile matter), and fractions of VV-mode particles were higher in warm months (especially in July) than in cold months, indicating that the coating on accumulation-mode soot particles was also stronger in warm months than in cold months. As previously mentioned, meteorological conditions in warm months (i.e., high *T* and RH) were favorable to the particle growth of soot particles through atmospheric photochemical and liquid chemical reactions. In cold months (May, September, and October), the volatility of accumulation-mode soot-containing particles was relatively lower, indicating thinner coating matter on the surfaces of soot particles in the polluted cold environment. This is consistent with measurements made at an urban site in Beijing (Yu et al., 2020). Yu et al. (2020) also suggests that a more even distribution of *r*BC and non-*r*BC material mass fractions in summer than in winter, which may be caused by higher amount of secondary material.

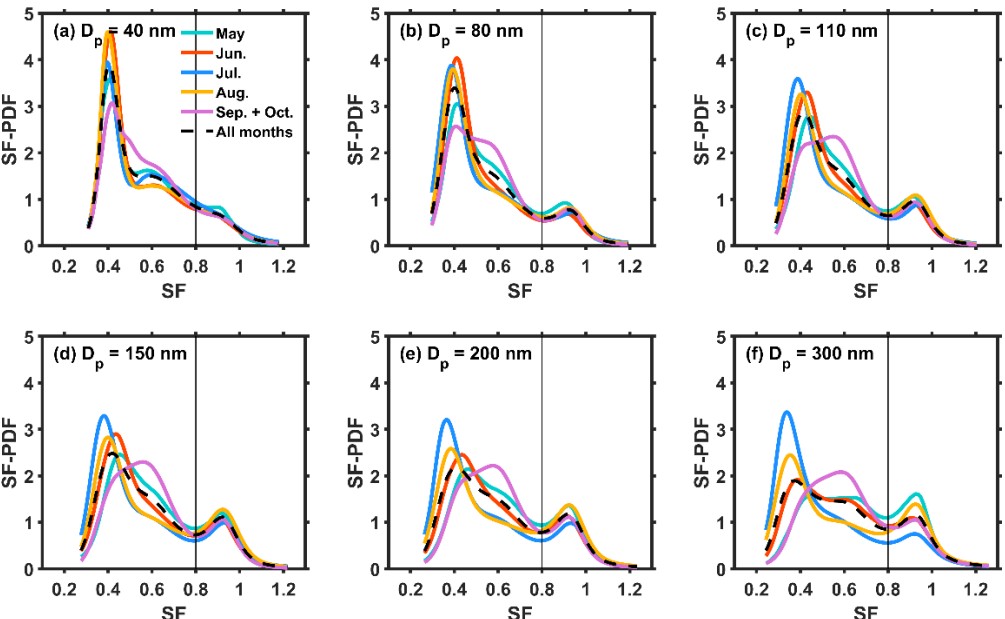

**Figure 4.** Monthly variations in the mean shrink factor (*SF*) probability distribution functions (*SF*-PDFs) for particles with diameters of (a) 40 nm, (b) 80 nm, (c) 110 nm, (d) 150 nm, (e) 200 nm, and (f) 300 nm.

Figure 5 shows diurnal variations in *SF*-PDF for different size particles, illustrating the distinct diurnal variation patterns of *SF*-PDF for nucleation- and accumulation-mode particles. VV-mode fractions for 40-nm and 80-nm particles (~*SF* = 0.4) increased sharply from around noon into the afternoon (Fig. 5a-b). Figure S3 shows that the number concentration of 40-nm and 80-nm particles increased quickly due to the influence of NPF events. This further corroborates that newly formed particles created during NPF events are the possible coating matter on nucleation-mode soot particles. Figure 5c-f suggests that NV-mode fractions in accumulation-mode soot particles (~*SF* = 0.9) were higher than those in nucleation-mode soot particles and that these fractions became higher with increasing particle size. NV-mode fractions in accumulation-mode soot particles clearly increased during the morning and evening rush hours. This suggests that anthropogenic emissions have a large impact on the volatility and mixing state of soot particles, especially for accumulation-mode soot particles. Previous studies have shown that some of the primary pollutants generated by human activities are composed of refractory materials, such as BC (Philippin et al., 2004; Levy et al., 2014). An increase in primary refractory particles could weaken the ensemble volatility and mixing state of soot particles. Figure 3c-f also shows that the NV-mode fraction in the *SF*-PDF of accumulation-mode particles decreased sharply in the daytime, likely caused by the coating effect of volatile matter through photochemical reactions.

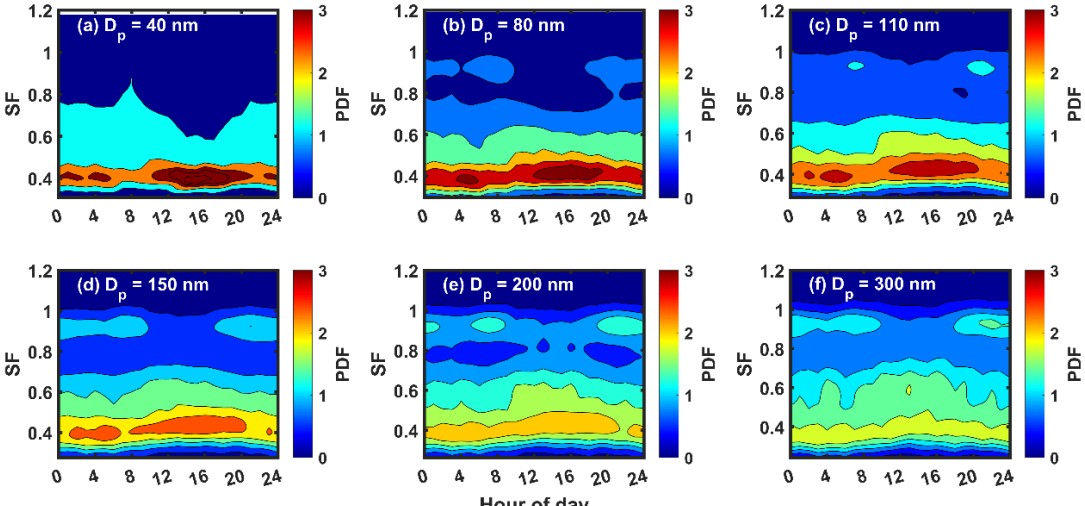

**Figure 5.** Diurnal variations in size-resolved shrink factor (*SF*) probability distribution functions (PDFs) for particles with diameters of (a) 40 nm, (b) 80 nm, (c) 110 nm, (d) 150 nm, (e) 200 nm, and (f) 300 nm.

In summary, the volatility and mixing state of soot-containing particles were complex at XT during the field campaign. Soot-containing particles in the nucleation mode had strong volatility and a high degree of internal mixing, likely due to the impact of frequent NPF events that occurred during this campaign. The strong volatility and high degree of internal mixing in warm months were likely caused by the aging processes of particles. Anthropogenic emissions also had a large impact on the volatility and mixing state of soot particles, especially in the accumulation mode. The impacts of anthropogenic emissions and secondary chemical reactions on the volatility and mixing state of soot particles will be further discussed next.

### 3.3 Factors influencing the volatility and mixing state of soot particles

#### 3.3.1 The impact of anthropogenic emissions on the volatility and mixing state of soot particles

As previously discussed, soot particles from anthropogenic emissions were always refractory and nonvolatile at 300°C. Analyzing the relationship between the number fraction of nonvolatile-mode particles ($NF_{NV}$, $SF > 0.8$) in *SF*-PDFs and $M_{BC}$ can verify this because BC is the main matter in soot particles. Figure 6a shows that $NF_{NV}$ reached two peak values, one during the morning rush hour at about 08:00 and the other during the evening rush hour at about 20:00. $M_{BC}$ also reached two peak values at those same times (Fig. 6b). Overall, the diurnal variation trends of $NF_{NV}$ for all sizes and $M_{BC}$ were similar. This suggests the great impact of anthropogenic BC on the volatility and mixing state of soot particles. $NF_{NV}$ decreased quickly after rush hours, especially in the

morning (Fig. 6a), suggesting that the aging processes of primary soot particles were quick at this heavily polluted site. Cheng et al. (2012) also observed the same phenomenon at a suburban site in Beijing.

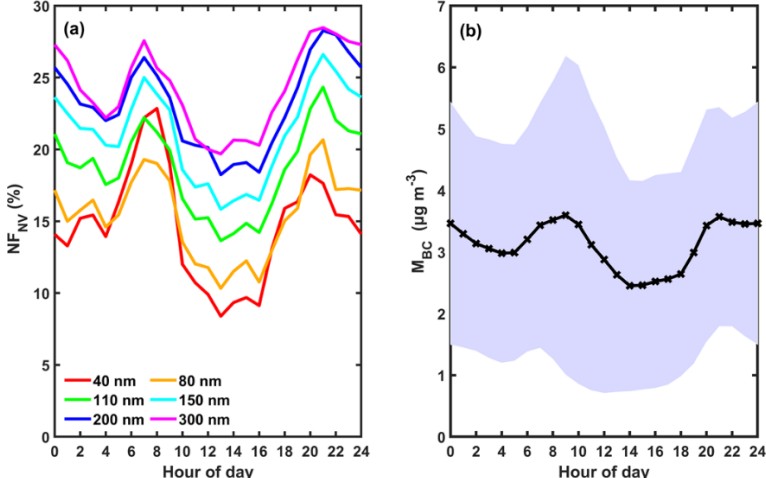

**Figure 6**. Diurnal variations in (**a**) wavelength-dependent, size-resolved number fractions of nonvolatile particles ($NF_{NV}$), and (**b**) mass concentration of black carbon ($M_{BC}$). The purple, shaded area shows the standard deviations of $M_{BC}$.

### 3.3.2 The impact of aging processes on the volatility and mixing state of soot-containing particles

Lower $SF_{mean}$ values mean stronger aerosol volatility, indicating a larger coating depth of volatile matter on soot particles. Figure 7a suggests that volatility is stronger during daytime than at night (i.e., a lower $SF_{mean}$), particularly for 40-nm particles. This illustrates the large impact of photochemical reactions on the volatility and mixing state of soot particles. Figure 7a also suggests that the $SF_{mean}$ of 80-nm particles was lower than that of 40-nm particles. Wang et al. (2018) suggests that aerosol hygroscopicity of 40-nm particles is larger than that of 80-nm particles during the daytime at this site. These indicate the great impact of photochemical reactions on the physicochemical properties of nucleation-mode particles. Inversely, $SF_{mean}$ increased with increasing particle size in the accumulation mode (110–300 nm), suggesting weaker volatility and a smaller coating depth for larger accumulation-mode soot particles.

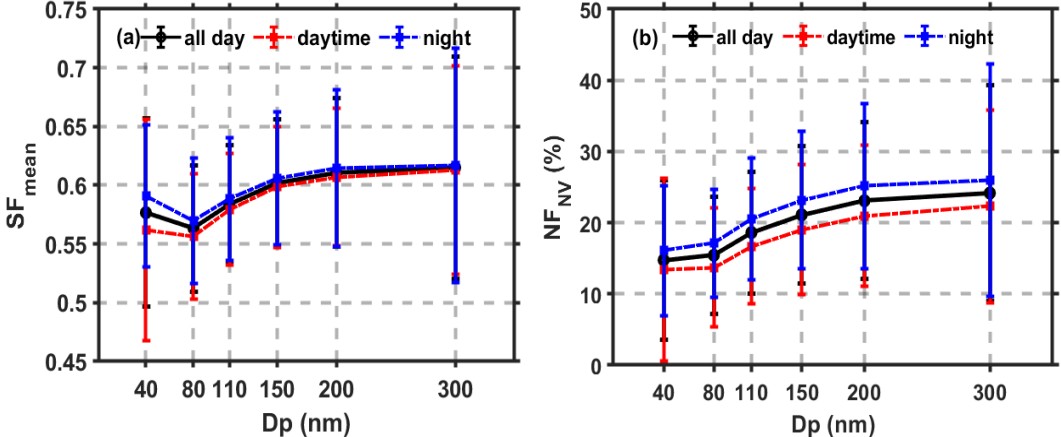

**Figure 7**. (**a**) Size-resolved ensemble mean shrink factors ($SF_{mean}$) and (**b**) size-resolved number fractions of nonvolatile particles ($NF_{NV}$) during the 24-hr day (black solid lines), during daytime (red dotted lines), and during nighttime (blue dotted lines). The error bars denote standard deviations.

Figure 8 shows the diurnal variation in $SF_{mean}$ in different months for different particle sizes. Figure 8a-b shows that the $SF_{mean}$ of 40-nm and 80-nm particles clearly increased during the morning and evening rush hours in all months. However, the $SF_{mean}$ of 40-nm and 80-nm particles decreased sharply in the afternoon. This suggests that the volatility of nucleation-mode soot-containing particles was easily influenced by anthropogenic emissions during rush hours and photochemical reactions in the daytime. The diurnal variation patterns of $SF_{mean}$ (Fig. 8c-f) in the accumulation mode were diverse in different months. The $SF_{mean}$ in warm months was usually lower than in cold months, indicating a larger impact of aging processes on the volatility of accumulation-mode soot-containing particles in warm months. Figure 8c-f also shows that the $SF_{mean}$ in accumulation mode was lowest in July. This suggests that high $T$, high RH, and the stable atmospheric environment in July were conducive to the coating of secondary matter on accumulation-mode soot particles, a possible reason for the high aerosol pollution levels in July. Moreover, Fig. 8 suggests that monthly variations in $SF_{mean}$ became larger with increasing particle size. The seasonal variation in the coating effect should thus be considered when modeling physicochemical properties of soot particles, especially larger particles.

To further investigate the impact of aging processes on the mixing state of soot particles, size-resolved $NF_{NV}$ in the daytime and at night were compared (Fig. 7b). $NF_{NV}$ was always lower in the daytime than at night, meaning that the fraction of externally mixed soot particles in the daytime was lower. This further indicates that photochemical reactions in the daytime can transform externally mixed soot particles into internally mixed soot particles. Figure 7b also shows that $NF_{NV}$ increased with increasing particle size, meaning a higher degree of external mixing of larger particles. This suggests that the degree of external mixing was higher for accumulation-mode soot particles than nucleation-mode particles.

The diurnal variation patterns of $NF_{NV}$ (Fig. S4) and $SF_{mean}$ (Fig. 8) in different months were
similar. Externally mixed soot particles increased during the morning and evening rush hours due
to enhanced anthropogenic emissions. Monthly differences in $NF_{NV}$ increased with increasing
particle size. Figure S4 also shows a lower number fraction of externally mixed soot particles (i.e.,
a smaller $NF_{NV}$) in warm months than in cold months.
These results illustrate the distinct volatilities and mixing states of soot particles between the
nucleation and accumulation modes. A lower degree of external mixing and thicker coating depth
in nucleation-mode particles exists. It is thus important to quantify the impact of the coating effect
for nucleation-mode soot particles when studying aerosol physicochemical properties. The next
section analyzes the coating depth and its influencing factors.

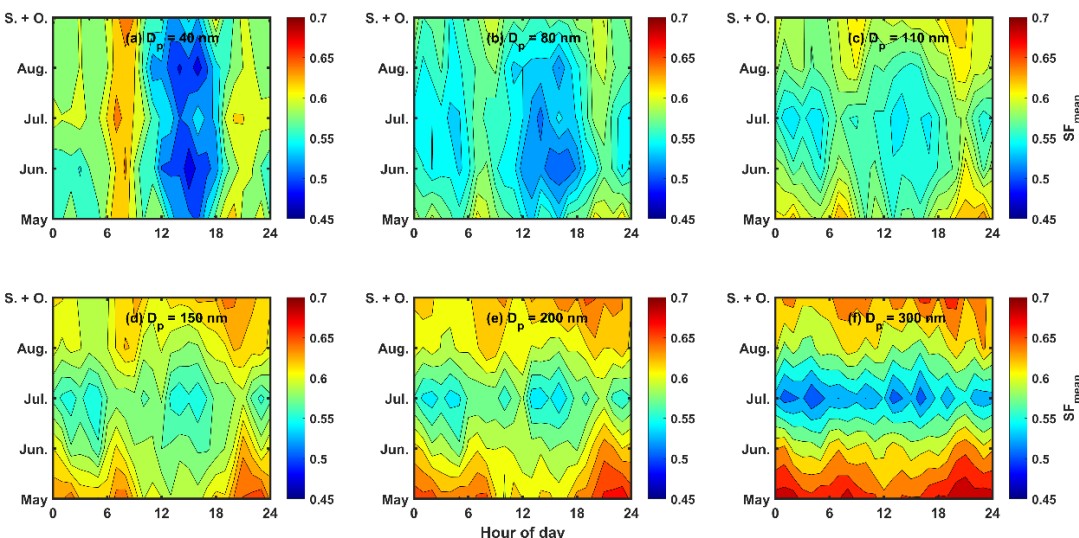

**Figure 8**. Diurnal variations in ensemble mean shrink factor ($SF_{mean}$) in different months for
different particle sizes.


3.4    The coating depth of secondary matter on soot particles

The ensemble mean coating depth on soot particles ($D_{c,mean}$) can be calculated using Eq. (4).
Figure 9 shows diurnal variations in $D_{c,mean}$ in different months for different particle sizes. The
diurnal variation patterns of $D_{c,mean}$ for nucleation-mode and accumulation-mode soot particles
differ greatly. The diurnal variation patterns of $D_{c,mean}$ in different months were similar for
nucleation-mode soot particles (40-nm and 80-nm particles) but not for accumulation-mode soot
particles (110–300-nm particles). The enhancement of $D_{c,mean}$ in the daytime occurred in all months
for nucleation-mode soot particles but only in the warm months for accumulation-mode soot
particles. At night, the enhancement of $D_{c,mean}$ for accumulation-mode soot particles was strong,
especially in warm months. However, it was weak for nucleation-mode soot particles. These all
imply large differences in $D_{c,mean}$ in different months for nucleation-mode and accumulation-mode
soot particles, likely caused by variations in meteorological conditions and aerosol pollution levels.

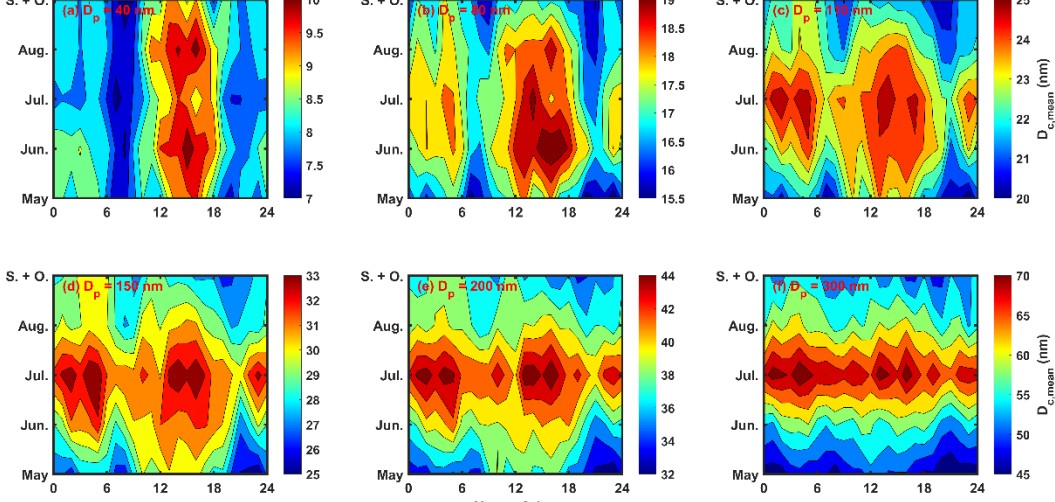


**Figure 9**. Diurnal variations in ensemble mean coating depth ($D_{c,mean}$) on soot particles in different
months for different particle sizes. Note that the color bars have different ranges of values in each
panel.

The relationships between $D_{c,mean}$ and several possible influencing factors ($T$, RH, and $PM_{10-}$
$_{400nm}$) were further analyzed (Fig. 10). Figures 10a and 10d show positive correlations between
$D_{c,mean}$ and $T$ for both nucleation-mode and accumulation-mode particles (represented by 40-nm and
300-nm particles, respectively). This is consistent with the results shown in Fig. 7. Zhang et al.
(2021) also indicated that warm environments were favorable to the aging of $r$BC. The high daytime
$T$ was conducive to the aging of soot particles caused by strong photochemical reactions. However,
the relationships between RH and $D_{c,mean}$ (Figs. 9b and 9e) and between $PM_{10-400nm}$ and $D_{c,mean}$ (Figs.
9c and 9f) were inverse between nucleation- and accumulation-mode soot particles.
Figure 9 depicts a linear relationship between $D_{c,mean}$ and RH, while a logarithmic relationship
between $D_{c,mean}$ and $PM_{10-400nm}$. $D_{c,mean}$ in the nucleation mode decreased with increasing RH and
$PM_{10-400nm}$ for nucleation-mode soot particles (Fig. 9b-c). This suggests that high ambient RH and
severe aerosol pollution events could inhibit the coating of nucleation-mode soot particles. Previous
studies have reported that aerosol pollution events were generally associated with high RH in the
NCP (G. Wang et al., 2016; Z. Wu et al., 2018). This suggests that highly polluted environments
with high ambient RH are not beneficial to the formation of new particles, leading to the weak
coating on nucleation-mode soot particles. However, $D_{c,mean}$ in the accumulation mode increased
with increasing RH and $PM_{10-400nm}$ (Fig. 9e-f). This suggests that highly polluted environments with
high ambient RH favor the growth of accumulation-mode soot particles by coating. This is possibly
related to enhanced liquid-phase chemical reactions under these environmental conditions.
Considering that accumulation-mode particles are the dominant components of $PM_{10-400nm}$, this
further implies that the coating on soot particles is important to the formation of heavy aerosol
pollution events. Y. Wang et al. (2019) indicated that the properties of ultrafine- and accumulation-
mode particles were distinct in clean and polluted urban environments due to the different particle
formation and growth processes. This study further indicates that it is also distinct in the aging of
soot particles.
In summary, high ambient $T$ and RH levels appeared to promote the coating growth of
accumulation-mode soot particles in highly polluted environments. High ambient $T$ but low RH
were beneficial to the coating growth of nucleation-mode soot particles in less polluted
environments.

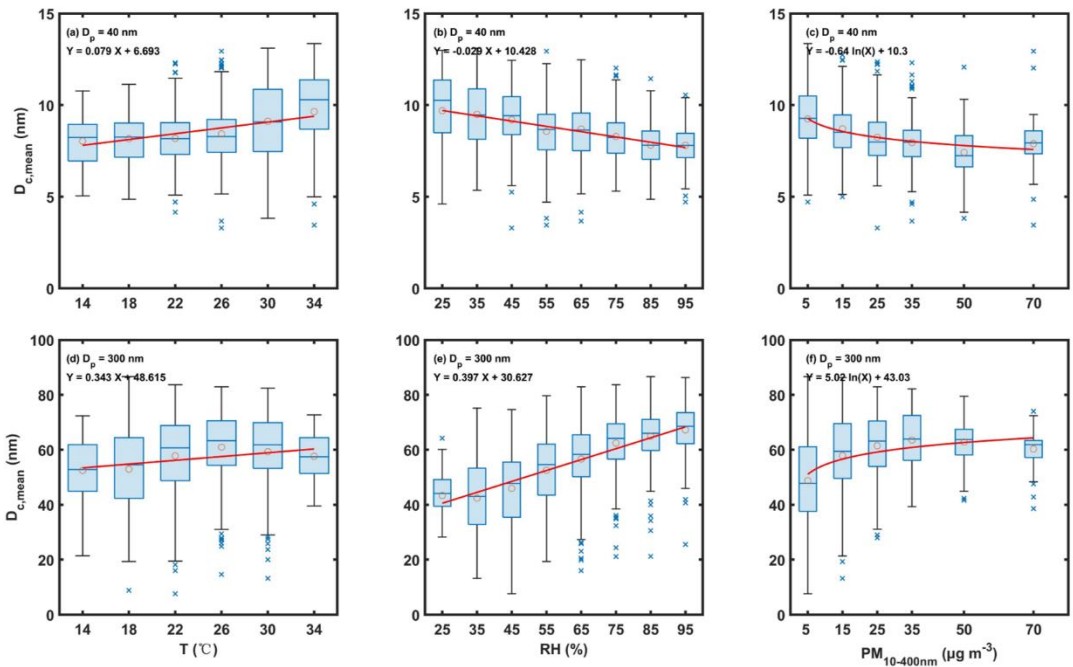

**Figure 10**. Relationships between ensemble mean coating depth ($D_{c,mean}$) and ambient $T$ (a, d) and
RH (b, e), and $PM_{10-400nm}$ (c, f) for 40-nm (top panels) and 300-nm (bottom panels) particles. The
circles show the mean $D_{c,mean}$ with boxes showing the 25th, 50th, and 75th percentiles and
extremities show the 5th and 95th percentiles. Red lines show the linear or logarithmic fitting lines
through the data, and best-fit relations are given in each panel.

## 4.     Summary and conclusions

Soot particles containing most of the black carbon (BC) in the atmosphere are the most
important light-absorbing carbonaceous particles. Investigating the mixing state of soot particles in

the field is crucial to accurately model aerosol absorption and reduce the uncertainty of radiative forcing caused by aerosols in climate models.

Here, over five months of volatility tandem differential mobility analyzer (VTDMA) data collected at a heavily polluted suburban site (Xingtai, or XT) from May to October of 2016 were used to study the volatility and mixing state of size-resolved soot particles and their influencing factors. Ambient meteorological variables [temperature ($T$), relative humidity (RH), and winds] varied between the warm (June, July, and August) and cold (May, September, and October) months of the field campaign. Variations in meteorological parameters could induce various aerosol aging processes and different levels of aerosol pollution, largely impacting the volatility and mixing state of soot particles.

The retrieved probability density function of the shrink factor ($SF$-PDF) at XT had three modes, demonstrating that the volatility and mixing state of soot-containing particles were more complex at XT than at other sites in the North China Plain. Compared with accumulation-mode soot-containing particles, nucleation-mode soot-containing particles were more volatile and had a higher degree of internal mixing. The diurnal variation patterns of $SF$-PDFs suggest that coating by newly formed materials was the possible reason for the enhanced volatility of nucleation-mode soot-containing particles in the daytime. Moreover, the enhanced nocturnal secondary aerosol formation was responsible for the enhanced volatility of accumulation-mode soot-containing particles in the nighttime. The ensemble mean $SF$ ($SF_{mean}$) was size dependent and varied monthly. The monthly variations in $SF_{mean}$ became larger with increasing particle size, implying a stronger seasonal variation of the coating effect on soot particles for larger-sized particles.

The similar diurnal variation trends of the number fraction of nonvolatile mode particles ($NF_{NV}$) in $SF$-PDFs and the mass concentration of BC ($M_{BC}$) suggest that human activities had a negative influence on the volatility and degree of internal mixing of soot particles, especially for accumulation-mode soot-containing particles. In general, less externally mixed soot particles (i.e., a smaller $NF_{NV}$) were present in warm months than in cold months. $NF_{NV}$ was always lower in the daytime than at night, suggesting a lower fraction of externally mixed soot particles in the daytime. This suggests that daytime photochemical reactions may promote the transformation of externally mixed soot particles into internally mixed soot particles. Moreover, $NF_{NV}$ increased with increasing particle size, meaning a higher degree of external mixing for larger-sized particles. This also suggests that the degree of external mixing was higher for accumulation-mode soot particles than for nucleation-mode soot particles.

To explore factors influencing soot-particle volatility and mixing state, the ensemble mean coating depth ($D_{c,mean}$) of volatile matter on soot particles was investigated. $D_{c,mean}$ was thicker in warm months than in cold months, even though aerosol pollution was heavier in cold months. In

warm months, $D_{c,mean}$ was larger in July than in other months, likely because high $T$, high RH, and the stable atmospheric environment in July were conducive to the coating effect on soot particles. The relationships between $D_{c,mean}$ and possible influencing factors (i.e., $T$, RH, and $PM_{10-400nm}$) show that high ambient $T$ and RH in a polluted environment promoted the coating growth of accumulation-mode soot particles. High ambient $T$ but low RH in a clean environment was beneficial to the coating growth of nucleation-mode soot particles.

These results demonstrate great differences in the volatility and mixing state between nucleation- and accumulation-mode soot particles. The impact of anthropogenic emissions on the volatility and mixing state of soot-containing particles was clearly seen, especially for accumulation-mode soot-containing particles. The monthly variations in meteorological conditions and aerosol pollution levels may induce different aerosol aging processes, strongly impacting the volatility and mixing state of soot-containing particles. This study suggests that differences between the mixing states of nucleation- and accumulation-mode soot particles and their influencing factors should be considered in climate models.

*Acknowledgement.* This work was funded by the National Natural Science Foundation of China (NSFC) research project (grant no. 42030606, 42005067, 92044303), the National Key R&D Program of the Ministry of Science and Technology, China (grant no. 2017YFC1501702), and the Open Fund of State Key Laboratory of Remote Sensing Science (grant no. 202015). We also thank all participants in the campaign for their tireless work and cooperation.

*Data availability.* The measurement data from the field experiment used in this study are available from the first author upon request (yuyingwang@nuist.edu.cn).

*Author contributions.* YW conceived the study and led the overall scientific questions. YW, RH, and QW processed the measurement data and prepared this paper. ZL, MC copyedited the article. Other co-authors participated in the implementation of this experiment and the discussion of this paper.

*Competing interests.* The authors declare that they have no conflict of interest.

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
