# Peer review of "Different effects of anthropogenic emissions and aging processes on the"

_Atmospheric Chemistry and Physics, 2021_

## Author Comment (AC1)

**Reply to RC1**

This paper shows some VTDMA data from China, with the intent of investigating factors affecting the mixing state of refractory material in a polluted environment. The results are interesting and within the remit of ACP, and the manuscript is reasonably well written. Also, because measurements of this nature are particularly common, there is an element of novelty in its own right. However, this paper is slightly let down by the fact that the results are interpreted in a very self-contained manner, without really considering the wider body of knowledge. Addressing this should be fairly straightforward, however this could potentially change the character of the paper, therefore I recommend publication after 'major' corrections.

**RE:** The reviewer raised an important issue. In the revised manuscript, we further highlight the novelty of this study following a more comprehensive review of more previous studies. Our study is unique in the analysis of the volatility and mixing state of nucleation-mode soot-containing particles. The findings help understand the aging processes of soot particles and improving the accuracy of modeled aerosol absorption. Meanwhile, we add more results to validate our results against previous findings in the revised manuscript.

Major comments:
The authors give an interesting discussion investigating the potential reasons for the phenomena they observe, however they do not place this in the context of wider atmospheric implications. In particular, this paper would benefit from a comparison with equivalent measurements in other locations, or alternative methods of measuring BC mixing state (e.g. doi: 10.5194/acp-20-3645-2020). This will allow for a deeper insight into the processes and phenomena under investigation.

**RE:** The reviewer provides a good reference for us to compare the two studies. Yu et al. (2020) characterize mass-resolved mixing state of mass-resolved black carbon (BC) in Beijing based on the measurement of a coupled combination of a centrifugal particle mass analyzer (CPMA) and a single-particle soot photometer (SP2). A new inversion algorithm was used to characterize the mixing state of rBC-containing particles in Beijing. We characterize the size-resolved mixing state of soot particles based on the measurement of VTDMA in Xingtai using a different observation method but the measurement result is comparable. For example, Yu et al. (2020) finds that a more even distribution of rBC and non-rBC material mass fractions in summer, which may be caused by higher amount of secondary material. In our study, we find that the coating effect of volatile matter on soot particles was stronger in warm months (i.e, in summer) than in cold months. Another example is that Yu et al. (2020) finds that polluted air from the Southern Plateau dominated the aged rBC-containing particles in Beijing. In our study, the measurement site (Xingtai) is in the south of Beijing where highly aged soot particles (less NV particles) were observed.

In addition, more comparisons are made with reference to other studies in the discussion.

It would also better justify this as an ACP research article (as opposed to a

measurement report) if either novel implications for wider atmospheric science could be specifically identified, or if newly-identified phenomena could be singled out.

RE: Most previous studies about mixing state of soot particles are based on the measurement of single-particle soot photometer (SP2) or soot particle aerosol mass spectrometer (SP-AMS). However, these measurements only denote the accumulation-mode soot particles. This is because the lower observation limit of particle size by SP2 and SP-AMS is larger than ~70 nm. VTDMA can make up this deficiency because its measurement is based on the aerosol number concentration, which is always high in the nucleation mode. This study firstly reports that the anthropogenic emissions and aging processes have different effects on the mixing state of nucleation- and accumulation-mode soot particles based on the five-month VTDMA measurement. Furthermore, factors influencing the coating depth of soot particles are found in this study. Some of the findings are new and important to improve the accuracy of modeled aerosol optical properties.

The discussion in the above has been added in the revised manuscript.

Minor comments:

Line 23: Taken in isolation, "weaken the volatility of soot particles" is a strange statement to make because many use the term "soot" synonymously with the refractory components like black carbon. I would rephrase.

RE: Agreed. It is revised as "weaken the mean volatility of soot-containing particles".

Page 69: Co-emitted organic carbon from biomass burning can be refractory (sometimes referred to as 'tar' or 'tarballs').

RE: Adachi et al. (2018, 2019) suggests that tarballs are mainly from biomass burning events (such as wildfires) and these particles are also refractory. The sentence is revised as "Aerosol volatility refers to the shrinking extent of particles at a certain temperature. The mixing state of soot particles or tarballs is closely related to aerosol volatility at high temperatures (Philippin et al., 2004; Wehner et al., 2009; Adachi et al., 2018, 2019).".

Line 157: This repeats a statement already made earlier.

RE: It is deleted.

Line 166: I presume the factory calibration was used to calculated MBC, but this should still be stated.

RE: Yes, it was. We have added the corresponding description in the revised manuscript.

Line 217: "better atmospheric diffusion conditions" needs to be better explained

RE: In August, there are more and stronger northerly winds, which is beneficial to the diffusion of air pollutants. The sentence is revised as "$PM_{10-400}$ was lower in August than in July, likely because of the better atmospheric diffusion conditions (more and stronger northerly winds) in August.".

Line 222: This paragraph doesn't really say anything substantial and can probably

be removed.

RE: This paragraph has been removed.

---

## Author Comment (AC2)

**Reply to RC2**

The authors present a five-month aerosol volatility measurement at a suburban site in North China Plain, focusing on analysis and interpretation of data from a volatility tandem differential mobility analyzer (VTDMA). The manuscript is like a measurement report. Throughout the manuscript, the authors tried to explain their measurements by some reasons that they could not demonstrate, using the sentences "likely caused by…"ï¼Œ"likely due to…"ï¼Œ"likely because of…." to interpret their results. Some discussions and conclusions are not reasonable and even wrong. For example, the authors conclude that anthropogenic emissions could weaken the volatility of soot particles and enhanced their degree of external mixing, which can not be supported by the measurement results that show increased fractions of non-volatile mode soot (i.e., externally mixed BC) and decreased coating depth with size increase. These measurement results follow the diffusion growth theory, namely condensation process of secondary aerosol components (i.e., coating materials) on soot surface is more sensitive to smaller size particles, in other word, the coating growth is effective for smaller soot particles.

RE: We appreciate the critical and constructive comments and have strived to address them to the capacity permitted by our measurements that are analyzed in more depth with reference to more previous studies concerning the mixing state of refractory BC in the NCP. Some conclusions of our study are consistent with previous studies. For example, Yu et al. (2020) found more evenly distributed rBC and non-rBC material mass fractions in summer, caused by higher amount of secondary material, while we find that the coating effect of volatile matter on soot particles was stronger in warm months (i.e, in summer) than in cold months.

As shown in Fig. 6a and Fig. S4 in the supplement, the fractions of non-volatile mode particles ($NF_{NV}$) increased obviously in the rush hours for both nucleation and accumulation mode particles. Figure 8 in the manuscript revealed that the ensemble mean shrink factor ($SF_{mean}$) increased in the rush hours. All these suggest that anthropogenic sources can weaken the mean volatility of soot-containing particles and enhance their degree of external mixing by emitting more fresh and externally mixed soot particles. Cheng et al. (2012) also observed the same phenomenon at a suburban site in Beijing. Wu et al. (2017) and Zhang et al. (2021) reported that refractory BC mass size distribution measurements in Beijing peaked at about 200 nm. This means that anthropogenic sources emitted more externally mixed soot particles in the accumulation mode than that in the nucleation mode.

The chemical processes on the surface of BC is complex, which is not only controlled by the diffusion growth theory. Recently, some new theories are put forward. For example, Zhang et al. (2020) demonstrates that BC-catalyzed sulfate formation involving $NO_2$ and $NH_3$ plays an important role in the particle growth and the development of hazes in China. This BC catalytic chemistry can occur even at low $SO_2$ and intermediate relative humidity levels. In addition, the coagulation with preexisting aerosols is also important for the coating of soot particles (e.g., He et al., 2015).

We added more references in the revised manuscript to confirm our measurements. Some

new findings are firstly reported in our study. For example, we find that the variation of meteorological conditions and pollution levels has different effect on the aging of the nucleation-mode and accumulation-mode soot particles based on the five-month VTDMA measurement.

**Reference:**

He, C., Liou, K. N., Takano, Y., Zhang, R., Levy Zamora, M., Yang, P., Li, Q., and Leung, L. R.: Variation of the radiative properties during black carbon aging: theoretical and experimental intercomparison, Atmos. Chem. Phys., 15, 11967-11980, 10.5194/acp-15-11967-2015, 2015.

Wu, Y., Wang, X., Tao, J., Huang, R., Tian, P., Cao, J., Zhang, L., Ho, K. F., Han, Z., and Zhang, R.: Size distribution and source of black carbon aerosol in urban Beijing during winter haze episodes, Atmos. Chem. Phys., 17, 7965-7975, https://doi.org/10.5194/acp-17-7965-2017, 2017.

Yu, C., Liu, D., Broda, K., Joshi, R., Olfert, J., Sun, Y., Fu, P., Coe, H., and Allan, J. D.: Characterising mass-resolved mixing state of black carbon in Beijing using a morphology-independent measurement method, Atmos. Chem. Phys., 20, 3645-3661, 10.5194/acp-20-3645-2020, 2020.

Zhang, F., Wang, Y., Peng, J., Chen, L., Sun, Y., Duan, L., Ge, X., Li, Y., Zhao, J., Liu, C., Zhang, X., Zhang, G., Pan, Y., Wang, Y., Zhang, A. L., Ji, Y., Wang, G., Hu, M., Molina, M. J., and Zhang, R.: An unexpected catalyst dominates formation and radiative forcing of regional haze, Proceedings of the National Academy of Sciences, 117, 3960, 10.1073/pnas.1919343117, 2020.

Zhang, Y., Liu, H., Lei, S., Xu, W., Tian, Y., Yao, W., Liu, X., Liao, Q., Li, J., Chen, C., Sun, Y., Fu, P., Xin, J., Cao, J., Pan, X., and Wang, Z.: Mixing state of refractory black carbon in fog and haze at rural sites in winter on the North China Plain, Atmos. Chem. Phys., 21, 17631-17648, 10.5194/acp-21-17631-2021, 2021.

General Comments:

The authors took 40-nm and 80-nm particles to explore volatility of nucleation-mode soot particles. The VTDMA measurement show that the particles in nucleation mode (represented by 40-nm particles) had strong volatility and a high degree of internal mixing with shrink factor of ~0.4, meaning that the residual particle size after heating was ~16 nm. These residual materials after heating at 300 degree for 40-nm particles are dominated by extremely low-volatile components rather than soot, taking into account that a carbon spherule of soot agglomerates has size of 15-30 nm. The authors claimed that the number concentration of 40-nm and 80-nm particles increased quickly due to the influence of new particle formation (NPF) events. Previous studies (e.g., Ehn et al., 2014) have demonstrated that the importance of extremely low-volatile organic components for the initial growth of new formed particles. These extremely low-volatile organic components remain in the particle phase after heating at 300 degree. How the authors to demonstrate that the residual materials after heating at 300 degree for 40-nm particles are soot rather than

extremely low-volatile organic components?

**RE:** The measurement site (Xingtai) is located in a highly polluted area in the central-south NCP because this region is heavily industrialized. Major industrial manufacturers include coal-based power plants, steel and iron works, glassworks, and cement mills (Wang et al., 2018). Wang et al. (2021) finds that both the mass concentration and mass fraction of BC in $PM_1$ are larger at Xingtai than those at urban Beijing in the north NCP. All these suggest that the emission of soot particles from the fossil fuel combustion is a lot at Xingtai.

The measurement of nucleation-mode soot particles is not easy because of their small volume and mass. Recently, Zhang et al. (2021) reports the size distribution of refractory BC in the mass equivalent diameter range of 70-500 nm based on the measurement of SP2 at the Gucheng site (between Beijing and Xingtai). It is found that the nucleation-mode soot particles are plentiful but they cannot be fully measured due to the limit of measurement size range by SP2. Considering that industrial emission at Xingtai is stronger than that at Gucheng, it is expected that more nucleation-mode soot particles are discharged at Xingtai. In addition, Fig. S4 in the supplement shows that the number fraction of nonvolatile particles ($NF_{NV}$) in the nucleation mode increased obviously during morning and evening rush hours, implying that traffic emission is also one of sources for nucleation-mode soot particles.

Ehn et al. (2014) demonstrates that extremely low-volatile components (ELVOCs) plays a considerable role in particle formation and growth in the forest area because forests emit large quantities of volatile organic compounds (VOCs). However, the primary aerosol sources are mainly from anthropogenic emissions in the NCP, leading to the heavy hazes in this region. The Xingtai site is located at the foothill of the Taihang Mountains. The weak diffusion conditions make it more easily influenced by the accumulation of air pollutants. The high concentrations of gaseous precursors and strong atmospheric oxidation capacity make new particle formation (NPF) occurring frequently at Xingtai (Wang et al., 2018; 2021).

Wehner et al. (2009) indicates that some nonvolatile material is produced during particle formation and growth in the polluted Beijing region, but usually ~97% of the particle material is volatile at 300 $^\circ$C. On the other hand, 97% of the newly formed particles consists of volatile particle material which is most likely dominated by sulfate but also volatile organic compounds. Cheng et al. (2012) also suggests that particles in the size range of 30-320 nm with non-volatile residuals at 300 $^\circ$C are mostly soot particles, which is measured at Yufa (another site in the NCP). In our previous studies, we found that the frequent NPF events are closely related to the formation of sulfate at Xingtai because this site is located in one of $SO_2$ pollution centers. (Wang et al., 2018, 2021).

According to the discussion above, we think most of nonvolatile particles in the nucleation mode are soot rather than ELVOCs although the contribution of ELVOCs is possible.

**Reference:**

Cheng, Y. F., Su, H., Rose, D., Gunthe, S. S., Berghof, M., Wehner, B., Achtert, P., Nowak, A., Takegawa, N., Kondo, Y., Shiraiwa, M., Gong, Y. G., Shao, M., Hu, M., Zhu, T., Zhang, Y. H., Carmichael, G. R., Wiedensohler, A., Andreae, M. O., and Pöschl, U.: Size-resolved measurement of the mixing state of soot in the megacity Beijing, China: diurnal cycle, aging

and parameterization, Atmos. Chem. Phys., 12, 4477-4491, 2012.

Ehn, M., Thornton, J. A., Kleist, E., Sipilä, M., Junninen, H., Pullinen, I., Springer, M., Rubach, F., Tillmann, R., Lee, B., Lopez-Hilfiker, F., Andres, S., Acir, I., Rissanen, M., Jokinen, T., Schobesberger, S., Kangasluoma, J., Kontkanen, J., Nieminen, T., Kurtén, T., Nielsen, L. B., Jørgensen, S., Kjaergaard, H. G., Canagaratna, M., Maso, M. D., Berndt, T., Petäjä, T., Wahner, A., Kerminen, V., Kulmala, M., Worsnop, D. R., Wildt, J., and Mentel, T. F.: A large source of low-volatility secondary organic aerosol, Nature, 506, 476-479, 10.1038/nature13032, 2014.

Wang, Y., Li, Z., Zhang, Y., Du, W., Zhang, F., Tan, H., Xu, H., Fan, T., Jin, X., Fan, X., Dong, Z., Wang, Q., and Sun, Y.: Characterization of aerosol hygroscopicity, mixing state, and CCN activity at a suburban site in the central North China Plain, Atmos. Chem. Phys., 18, 11739-11752, 10.5194/acp-18-11739-2018, 2018.

Wang, Y., Wang, J., Li, Z., Jin, X., Sun, Y., Cribb, M., Ren, R., Lv, M., Wang, Q., Gao, Y., Hu, R., Shang, Y., and Gong, W.: Contrasting aerosol growth potential in the northern and central-southern regions of the North China Plain: Implications for combating regional pollution, Atmos. Environ., 267, 118723, https://doi.org/10.1016/j.atmosenv.2021.118723, 2021.

Wehner, B., Berghof, M., Cheng, Y. F., Achtert, P., Birmili, W., Nowak, A., Wiedensohler, A., Garland, R. M., Pöschl, U., and Hu, M.: Mixing state of nonvolatile aerosol particle fractions and comparison with light absorption in the polluted Beijing region, Journal of Geophysical Research Atmospheres, 114, 85-86, 2009.

Zhang, Y., Liu, H., Lei, S., Xu, W., Tian, Y., Yao, W., Liu, X., Liao, Q., Li, J., Chen, C., Sun, Y., Fu, P., Xin, J., Cao, J., Pan, X., and Wang, Z.: Mixing state of refractory black carbon in fog and haze at rural sites in winter on the North China Plain, Atmos. Chem. Phys., 21, 17631-17648, 10.5194/acp-21-17631-2021, 2021.

Specific comments:

Abstract: I don't think the May, September and October with temperatures in the range of 15-30 degrees are cold months.

RE: As mentioned in the abstract, warm months (June, July and August) and cold months (May, September and October) are relative to emphasize the difference of ambient temperature in these months. Actually, other meteorological variables are also distinct between the cold and warm months, such as the ambient RH shown in Fig. 2a, and the winds shown in Fig.2b and Fig. S1. This is because the weather system affecting the measurement site differ markedly between the warm and cold months. For example, this site is influenced periodically by strong cold fronts in cold months but not in warm months. Therefore, the meteorological variables changed periodically in cold months but not in warm months. The difference of meteorological variables can cause the variation of aerosol volatility and soot mixing state, as discussed in this paper. In addition, Fig. 2c shows that the heavy haze is more likely to appear in cold months.

Introduction (Page 3/Lines 61-62): The authors stated that "However, studies on the mixing state of BC or soot particles in the actual atmosphere are few due to limited observations." To my knowledge, there have been many field measurements to investigate the mixing state of ambient BC particles using a

single-particle soot photometer (SP2) and a soot particle aerosol mass spectrometer (SP-AMS) in recent years.

**RE:** The sentence "However···" is deleted. The paragraph about the measurement results using SP2 and SP-AMS in the NCP was added.

"The online measurement instruments quantifying the mixing state of BC-containing particles are limited. Based on the measurement of single-particle soot photometer (SP2), Wu et al. (2017) indicated that the mass of refractory black carbon (rBC) had an approximately lognormal distribution as a function of the volume-equivalent diameter (VED) in Beijing. Yu et al. (2020) suggested that the mixing state of rBC particles was related to air pollution levels and air mass sources. Zhang et al. (2021) further indicated that meteorological conditions had a large impact on the mixing state of rBC particles. Moreover, the Aerodyne soot particle aerosol mass spectrometer (SP-AMS) can also be used to study the mixing state of rBC. For example, J. Wang et al. (2019) found that the formation of secondary aerosols through photochemical and aqueous chemical reactions was responsible to the coating of rBC based on the measurement of SP-AMS in winter Beijing. However, the lower measurement size limit of SP2 and SP-AMS is larger than 70 nm. Therefore, they cannot quantify the mixing state of BC-containing particles in the small nucleation mode.".

Thanks for the suggestion.

2.2 Measuring BC: The raw data of AE33 measurement have a large uncertainty due to filter-loading and multiple-scattering effects. However, the authors did not make any corrections for measurement data.

**RE:** Sorry, we didn't elaborate the correction but we did it. Drinovec et al. (2015) reported that a new real-time loading effect compensation algorithm is used in AE33 based on a two parallel spot (i.e., dual-spot) measurement of optical absorption. The dual-spot compensation algorithm determines the value of the compensation parameter $k$ with high temporal resolution, which indicates changes in aerosol properties on the daily timescale. Intercomparison studies show excellent reproducibility of the AE33 measurements and very good agreement with post-processed data obtained using earlier Aethalometer models and other filter based absorption photometers. A wavelength-independent multiple-scattering compensation factor (2.90) adapted in eastern China recommended by Zhao et al. (2020) is used to process the data of AE33.

**Reference:**

Drinovec, L., Močnik, G., Zotter, P., Prévôt, A. S. H., Ruckstuhl, C., Coz, E., Rupakheti, M., Sciare, J., Müller, T., Wiedensohler, A., and Hansen, A. D. A.: The "dual-spot" Aethalometer: an improved measurement of aerosol black carbon with real-time loading compensation, Atmos. Meas. Tech., 8, 1965-1979, 10.5194/amt-8-1965-2015, 2015.

Zhao, G., Yu, Y., Tian, P., Li, J., Guo, S., and Zhao, C.: Evaluation and Correction of the Ambient Particle Spectral Light Absorption Measured Using a Filter-based Aethalometer, Aerosol Air Qual. Res., 20, 1833-1841, 10.4209/aaqr.2019.10.0500, 2020.

Page 7/Line 188:Average temperatures in warm (June, July, and August) and

cold (May, September, and October) months only have a difference of ~6 degree. The authors should reconsider classification criteria to discuss monthly variations.

RE: As described in the above, the warm and cold months are relative terms. The difference of meteorological conditions made different air pollution levels at Xingtai, leading to distinct aerosol properties between warm and cold months.

Page 8/Lines 218-219: The similar changes in concentrations of BC and PM can not suggest the non-trivial role of BC in the formation processes of aerosol pollution. Both concentrations of BC and PM strongly depend on planetary boundary layer height.

RE: We can't confirm the role of BC in the formation processes of aerosol pollution in this study, but a recent study demonstrated that BC catalyzed sulfate formation involving $NO_2$ and $NH_3$ may play an important role in the formation of haze events in China (Zhang et al., 2020). For this reason, this sentence is revised as: "Figure 2c also shows that changes in $M_{BC}$ and PM10-400 were similar, suggesting the **possible** role of BC in the formation processes of aerosol pollution.".

**Reference:**

Zhang, F., Wang, Y., Peng, J., Chen, L., Sun, Y., Duan, L., Ge, X., Li, Y., Zhao, J., Liu, C., Zhang, X., Zhang, G., Pan, Y., Wang, Y., Zhang, A. L., Ji, Y., Wang, G., Hu, M., Molina, M. J., and Zhang, R.: An unexpected catalyst dominates formation and radiative forcing of regional haze, Proceedings of the National Academy of Sciences, 117, 3960, 10.1073/pnas.1919343117, 2020.

Page 9/Lines 246: should be "non-volatile"

RE: It is revised. Thanks for the detailed check.

Page 12/Lines 326-328: There are only two sizes in nucleation mode. The authors made conclusion that the volatility of nucleation-mode soot particles became larger with increasing particle size, which is not solid.

RE: The sentence is deleted.

Figure 10: The coating depth and temperature have a poor correlation (R2=0.03-0.04), which can not support that coating depth depends on temperature as discussed by the authors.

RE: There are multiple factors influencing the coating depth of soot particles ($D_{c,mean}$), leading to a weak correlation when analyzing the relationship of $D_{c,mean}$ with one of these factors. In the revised manuscript, we analyzed the relationships using the box plots instead of scatter plots and fit the relationship of $D_{c,mean}$ and PM$_{10-400nm}$ using the logarithmic function instead of linear function. The results are shown in the figure below, showing the better relationships with these factors. We added more discussions about their relationships in this section.

[Figure]

Figure R1. Relationships between ensemble mean coating depth ($D_{c,mean}$) and ambient T (a, d) and RH (b, e), and PM$_{10-400nm}$ (c, f) for 40-nm (top panels) and 300-nm (bottom panels) particles. The circles show the mean $D_{c,mean}$ with boxes showing the 25th, 50th, and 75th percentiles and extremities show the 5th and 95th percentiles. Red lines show the linear or logarithmic fitting lines through the data, and best-fit relations are given in each panel.

**Reference:**

Wang, Y., Li, Z., Zhang, R., Jin, X., Xu, W., Fan, X., Wu, H., Zhang, F., Sun, Y., Wang, Q., Cribb, M., and Hu, D.: Distinct Ultrafine- and Accumulation-Mode Particle Properties in Clean and Polluted Urban Environments, Geophys. Res. Lett., 46, 10918-10925, 10.1029/2019GL084047, 2019.

Zhang, Y., Liu, H., Lei, S., Xu, W., Tian, Y., Yao, W., Liu, X., Liao, Q., Li, J., Chen, C., Sun, Y., Fu, P., Xin, J., Cao, J., Pan, X., and Wang, Z.: Mixing state of refractory black carbon in fog and haze at rural sites in winter on the North China Plain, Atmos. Chem. Phys., 21, 17631-17648, 10.5194/acp-21-17631-2021, 2021.

Page 17/Lines 441-442: Which measurement results can demonstrate enhanced nocturnal liquid chemical reactions?

**RE:** This sentence is revised as: "Moreover, the enhanced nocturnal secondary aerosol formation was responsible for the enhanced volatility of accumulation-mode soot particles in the nighttime.". Zhang et al. (2018) reported that the nocturnal chemical processes made the obvious increase of secondary aerosols such as nitrate at Xingtai.

**Reference:**

Zhang, Y., Du, W., Wang, Y., Wang, Q., Wang, H., Zheng, H., Zhang, F., Shi, H., Bian, Y., Han,

Y., Fu, P., Canonaco, F., Prévôt, A. S. H., Zhu, T., Wang, P., Li, Z., and Sun, Y.: Aerosol chemistry and particle growth events at an urban downwind site in North China Plain, Atmos. Chem. Phys., 18, 14637–14651, 10.5194/acp-18-14637-2018, 2018.

---

## Author Comment (AC3)

**Reply to RC3**

The manuscript investigates the daily and seasonal variability of soot particle mixing state coupling black carbon measurements and volatility tandem differential mobility analyzer data collected in a suburban site of the North China Plain (NCP).

The introduction reports that several other studies investigated the mixing state of soot particles using volatility measurements in the NCP. The introductions underlines that the present study differs from the previous ones because it encompasses two different seasons during a 5-month period. It is important to highlight what are the novelty of the results of this study compared to the previous ones, thanks to the multiple season measurements.

**RE:** In the revised manuscript, we further highlight the novelty of this study in the introduction. Most previous studies about mixing state of soot particles are based on the measurement of single-particle soot photometer (SP2) or soot particle aerosol mass spectrometer (SP-AMS). However, these measurement results only denote the accumulation-mode soot particles. This is because the lower measurement size limit of SP2 and SP-AMS is larger than 70 nm. VTDMA can make up this deficiency because its measurement is based on the aerosol number concentration, which is always high in the nucleation mode. This study for the first time reports that the anthropogenic emissions and aging processes have different effects on the mixing state of nucleation- and accumulation-mode soot particles. Thanks to the five-month VTDMA measurement, factors influencing the coating depth of soot particles are found and their relationships are established in this study. All findings are beneficial to study the aging processes of soot particles and improve the accuracy of modeled aerosol optical properties.

Some of the conclusions are not supported adequately by observations. Figure 4 shows that VV nucleation particles are characterized by a higher volatility during warmer months. On the other hand, the conclusion concerning the seasonal variability of nucleation mode soot particles relies on the assumption that nucleation mode soot is totally internally mixed. This assumption is not adequately supported by the presented results. For example, Figure 3 shows that nucleation mode particles are characterized by a multimodal distribution of SF, and soot could be responsible for the SV and NV peaks, which do not present the temperature trend discussed by the authors (higher values in warmer months). A deeper discussion of the results is encouraged. At line 441 the authors state that "Moreover, enhanced nocturnal liquid chemical reactions were responsible for the enhanced volatility of accumulation-mode soot particles in the nighttime." No clear evidence of liquid or heterogeneous phase reactions during night-time is provided in this study to support such a statement. Furthermore, soot particle coating is controlled by condensation of vapor phase compounds and coagulation with other particles (Bond et al., 2013; Ko et al., 2020). The link between soot coating and NPF events is quite speculative and is not clear (line 274). If the author are interested in investigating such a link, the particle number in the range 10 -100 nm should be investigated, rather than solely the change in

particle number concentration at 40 nm and 80 nm, as done at line 274.

**RE:** In the conclusions, we suggest that nucleation-mode soot particles were more volatile and had a higher degree of internal mixing than accumulation-mode soot particles, but the soot particles in any modes are not fully internally mixed in our measurement. Figures 6a and S4 suggest that the number fraction of nonvolatile particles ($NF_{NV}$) in nucleation-mode particles were always smaller than those in accumulation-mode particles but they are always larger than 0.

The line 441 sentence has been revised as: "Moreover, the enhanced nocturnal secondary aerosol formation was responsible for the enhanced volatility of accumulation-mode soot particles in the nighttime."In our previous study (Zhang et al., 2018), we found that the nocturnal chemical processes made the obvious increase of secondary aerosols such as nitrate at Xingtai.

The figure below shows the time series of the total number concentration of 10-100 nm particles ($N_{10-100\ nm}$). It suggests $N_{10-100\ nm}$ increased sharply on many days, indicating the frequent occurrence of NPF events. These newly formed particles should have an important impact on the growth of particles. The diurnal variation of $N_{10-100\ nm}$ is added in the Fig. S3 in the supplement.

[Figure]

**Reference:**

Zhang, Y., Du, W., Wang, Y., Wang, Q., Wang, H., Zheng, H., Zhang, F., Shi, H., Bian, Y., Han, Y., Fu, P., Canonaco, F., Prévôt, A. S. H., Zhu, T., Wang, P., Li, Z., and Sun, Y.: Aerosol chemistry and particle growth events at an urban downwind site in North China Plain, Atmos. Chem. Phys., 18, 14637-14651, 10.5194/acp-18-14637-2018, 2018.

Specific comments:

Line 166. Please specify if BC concentration was retrieved using the MAC suggested by the manufacturer or a site-specific MAC. In addition, the fact that BC is retrieved form optical measurements and the dependency of MAC on BC coating introduce some limitations in discussing BC concentration variability. The authors should mention this limitation in the discussion of results.

**RE:** The sentence is added in section 2.2.2 "According to the manufacturer's instructions, the MAC is calculated from the change in optical attenuation at channel 6 (i.e., 880 nm) in the selected time interval using the mass absorption cross section (MAC) of 7.77 $m^2\ g^{-1}$. The dependency of MAC on BC coating may introduce some uncertain in calculating MAC (Drinovec et al., 2015)."

**Reference:**

Drinovec, L., Močnik, G., Zotter, P., Prévôt, A. S. H., Ruckstuhl, C., Coz, E., Rupakheti, M., Sciare,

J., Müller, T., Wiedensohler, A., and Hansen, A. D. A.: The "dual-spot" Aethalometer: an improved measurement of aerosol black carbon with real-time loading compensation, Atmos. Meas. Tech., 8, 1965-1979, 10.5194/amt-8-1965-2015, 2015.

Line 228: It is not clear what the authors mean when they report the wavelength dependent size resolved SF-PDF. It looks like figure 3 reports the SF-PDF for different particle sizes, as selected by DMA1.
**RE:** It is a mistake. The words "wavelength dependent" are deleted. Thanks for the check.

Line 239 – 240: I suggest the authors to be more accurate, indicating that previous studies observed fresh BC in the lower bound of the accumulation mode. In fact, Levy et al. 2014 reports that the highest frequency of externally mixed fresh BC is observed at 150 nm, while Wu et al., 2017 reported that rBC size distribution measurements in Beijing peaked at about 200 nm, with a secondary less significant mode at 600 nm.
**RE:** Thanks for the suggestion. The sentence is revised as:"Some studies suggest that freshly emitted refractory particles (like BC) are primarily in accumulation mode. For example, Levy et al. (2013) reported that fresh BC was mostly in the 150–240 nm size range, while Wu et al. (2017) reported that refractory BC size distribution measurements in Beijing peaked at about 200 nm, with a secondary less significant mode at about 600 nm."

Paragraph 3.3.2 The statistical significance of the differences between day-time and night time observed in Figure 7 looks small. Could the authors comment on the observed variability?
**RE:** In Fig. 7, the daytime hours are 07:00~19:00, and the nighttime to 19:00~07:00. Figure S4 in the supplement suggests that NV mode particles increased obviously during the morning and evening rush hours, which influences the comparison results of $SF_{mean}$ and $NF_{NV}$ between the daytime and nighttime. The observed variability is mainly caused by the data in rush hours.

Figure 10: Did the author explore a different type of fitting for the relationship between coating thickness (Dc) and PM$_{10-400}$ (for example a logarithmic fitting rather than a linear fitting)?
**RE:** That is a good suggestion. The new fitting figures are shown below, showing a logarithmic relationship between $D_c$ and PM$_{10-400}$. We add more discussion in this section.

[Figure]

Technical corrections:

Line 71-72. This statement is true in polluted environments

**RE:** This sentence has been revised to "This is why aerosol volatility can characterize the mixing state of soot particles in polluted environments".

Line 115. A list of the measured meteorological parameters should be added to complete this sentence.

**RE:** The sentence is added "The measured meteorological variables including ambient temperature, relative humidity (RH), wind direction and speed was used in this study.".

Line 137: The scope of DMA1 is to select particles with a specific mobility dimeter, thus it would be more accurate to write: "the water-based condensation particle counter (WCPC, model 3787, TSI Inc.), measuring the number of particles ranging from 10 to 400 nm.

**RE:** It is revised. Thanks.

Line 192: From Figure 2 it looks like in July wind from southeast was present instead of prevalent. Please revise the sentence accordingly.

**RE:** The sentence is revised as "In July, weak southeast winds were also present, beneficial to the accumulation of air pollutants due to the stable atmospheric environment.".

Line 194: please specify when wind speed is considered high. From figure 2 it is difficult to understand if winds in August were stronger than in the other months.

**RE:** It is not appropriate. The words "always strong" are deleted. The sentence is revised as "In August, the other prevailing wind was from the north, which was beneficial for atmospheric diffusion.".

Line 204: please add a reference for the assumed particle density.
RE: The reference of Y. Wang et al. (2017) is added.

Line 439: coating of soot takes place for condensation of newly formed material and not newly formed particles.
RE: It is revised. Thanks.

References:

Ko et al., Atmos. Chem. Phys., 20, 15635–15664, 2020
Bond et al., J. Geophys. Res.-Atmos., 118, 5380–5552, 2013

---

## Editor Decision (ED1)

manuscript text in black.

Modern gasoline direct injection (GDI) vehicles can emit plentiful ultrafine BC  containing (soot) particles in the ambient (La Rocca et al., 2015; Hu et al., 2021).

The paragraph is about mixing state of soot particle. However, the information on size range of BC particles should come first, after the sentence in that starts in l. 58 (Black carbon is the most...). Please add not only the references by La Rocca 2015, and Hu et al., 2021 but also a reference that reports on BC in larger particles. For example, you could add briefly some information you wrote in the author response to the first editor comment.

  The tiny soot   particles embedded in other material (such as sulfate) play a significant role in particle growth (Li  et al., 2011).

This sentence implies that the soot itself leads to particle growth. This is not reflected in the article by Li et al., 2011. They only say that particle growth occurs on particles that contain soot – among other components. Please reword it or remove sentence.

Investigating the mixing state of BC-containing particles and their factors in different modes are needed.

This sentence seems out of place or redundant after the preceding paragraph where the role of mixing state is already discussed.

2) l. 258: These tiny soot particles in nucleation mode are mainly from modern vehicle emissions (La Rocca et al., 2015; Hu et al., 2021)

---

## Author Response (AR2)

We thank the reviewer for providing insightful comments and helpful suggestions that have substantially improved the manuscript. Below we have included the review comments in black followed by our responses in blue. In the revision of this manuscript, we have highlighted those changes accordingly.

I would like to thank the authors for addressing most of my minor comments. Nevertheless there are still some major comments that have not been adequately considered.

Lines 281-283 read "Figure 4a-b shows that VV-mode fractions in the SF-PDFs of 40-nm and 80-nm particles were higher in warm months than in cold months, indicating that nucleation-mode soot particles were more volatile in warm months"

The figure clearly shows that the fraction of very volatile nucleation mode particles was higher in the warmer months, but the authors should explain how they concluded that the very volatile particles in 40 and 80 nm range contain soot.

RE: The phenomenon of soot formation can be essentially described in terms of three steps: nucleation, growth and oxidation (Clague et al., 1999). The diameter of initial soot particles is lower than 2 nm. Surface growth, coagulation and aggregation make soot particles grow to tens or hundreds of nm. One method to measure soot particles is using the transmission electron microscopy (TEM). The TEM observations suggest that modern gasoline direct injection (GDI) can emit plentiful soot particles with a diameter smaller than 100 nm (e.g., La Rocca et al., 2015; Amin et al., 2019; Hu et al., 2021). For example, La Rocca et al. (2015) reported that primary soot particles from the emission of light-duty EURO IV GDI engine presented an average diameter of 36 nm with a mode of 32 nm (shown in the first figure below). Hu et al. (2021) sampled particle matter emitted from a China VI vehicle over the Worldwide harmonized Light vehicles Test Cycles (WLTC) and they found that ultrafine soot particles in diameter below 23 nm take up majority of particle emissions (shown in the second figure below). All this reflects that there are many ultrafine soot particles emitted from modern vehicles. Zhang et al. (2021) presented the size distribution of refractory black carbon ($r$BC) above 70 nm measured by SP2 at a site in the NCP (shown in the third figure below). From the figure, it is expected that there are many $r$BC below 70 nm in the ambient although the measurement is limited by the measurement size range of SP2.

[Figure]

The measured soot particles diameter distribution from light-duty EURO IV GDI engine (La Rocca et al., 2015)

[Figure]

The measured soot particle diameter distribution in different oxidation temperature from China VI GDI engine (Hu et al., 2021)

[Figure]

Time series of the mass size distribution and number size distribution of rBC, as measured by the SP2 at the Gucheng site in the NCP (Zhang et al., 2021).

Line 283-287. The authors reported that NPF events are frequent at the site. The growth of new particles to the size range of 40 and 80 nm is controlled by condensation. Nevertheless, condensation on newly formed particles would be in competition with condensation on soot particles (Matsui et al., 2021). The sentence "All this implies that coating by newly formed secondary matter was the possible reason for the high volatility of nucleation-mode soot-containing particles in warm months. " does not seem correct. Again, the authors need to demonstrate that the very volatile nucleation mode particles contain soot before discussing the high volatility of nucleation-mode soot particles. RE: NPF events occurred frequently in the North China Plain due to the high concentration of gaseous precursors and strong atmospheric oxidation capacity (Wang et al., 2017; Wang et al., 2018). Secondary matters such as sulfate were produced during NPF events. The mixing pathways of soot with other material are various, including condensation, coagulation and so on. The newly formed particles during events are easily captured by soot particles due to their irregular shapes. Li et al.

(2011) indicated that the tiny soot particles embedded in sulfates could promote particle growth during NPF events, which is based on the TEM observations in the NCP.

Finally, particles smaller than 50 nm after removal of condensed material are less likely to be soot, because primary soot particles are generally larger than 50 nm (Harris 2001). On the other hand, smaller particle containing very volatile molecules are likely composed by non-volatile organics (Kalberer et al., 2004; Wehner et al., 2009). The high mass concentration of BC at the site is not enough to demonstrate that the non-volatile material observed at the site is composed by soot.

RE: Harris and Maricq (2001) reported the size distribution for diesel and gasoline engine exhaust particulate matter from different engine classes: diesel, direct injection gasline, and port fuel injection gasoline. Sine then the engine has been greatly upgraded. Hu et al. (2021) indicated that modern gasoline direct injection (GDI) offered higher power output, improved fuel economy and reduced $CO_2$ emissions compared with port fuel injection (PFI) vehicles. However, the particle number emission of GDI is larger than that of PFI and also higher than diesel vehicles equipped with diesel particle filter (DPF). The GDI produced more ultrafine soot particles shown in the figures above. Kalberer et al. (2004) characterized the volatility of organics based on the measurement at the maximum temperature of 200ºC. However, 200ºC is too low to vaporize some volatile or semi-volatile material. Thermo-desorption at 250 to 300ºC appeared to be the optimum temperature to avoid size dependent effect due to limited residence time in the thermo-desorption unit (Villani et al., 2007). In our measurement, the temperature of VTDMA was set at 300ºC. Wehner et al. (2009) indicated that volatile compounds such as sulfates, nitrates, and most of organics species are evaporated at 300ºC. Residual particles are either externally or internally mixed BC particles or other nonvolatile material such as sea salt or crustal particles. In heavily polluted areas and for the submicrometer range, the majority of the nonvolatile particle mass can be assumed to be soot. Wehner et al. (2009) also suggested that some nonvolatile material is produced during particle formation and growth in the polluted Beijing region, but usually ~97% of the particle material is volatile at 300ºC. On the other hand, 97% of the newly formed particles consists of volatile particle material which is most likely dominated by sulfate but also volatile organic compounds.

Figure 6a and Fig. S4 in our paper suggest that the fractions of non-volatile mode particles ($NF_{NV}$) increased obviously in the rush hours for ultrafine particles, which can indirectly indicate that a large amount of ultrafine soot particles are from vehicle emissions.

References:

Matsui et al., 2021, Impact of new particle formation on the concentrations of aerosols and cloud condensation nuclei around Beijing, https://doi.org/10.1029/2011JD016025

Harris, J. S., and M. M. Maricq (2001), Signature size distributions for diesel and gasoline engine exhaust particulate matter, J. Aerosol Sci., 32, 749–764.

Kalberer, M., et al. (2004), Identification of polymers as major components of atmospheric organic aerosols, Science, 303, 1659– 1662.

Wehner, B., Berghof, M., Cheng, Y. F., Achtert, P., Birmili, W., Nowak, A., Wiedensohler, A.,

Garland, R. M., Pöschl, U., and Hu, M.: Mixing state of nonvolatile aerosol particle fractions and comparison with light absorption in the polluted Beijing region, J. Geophys. Res. Atmos., 114, 85-86, 2009.

**Reference:**

Amin, H. M. F., Bennett, A., and Roberts, W. L.: Determining fractal properties of soot aggregates and primary particle size distribution in counterflow flames up to 10 atm, P. Combust. Inst., 37, 1161-1168, https://doi.org/10.1016/j.proci.2018.07.057, 2019.

Clague, A. D. H., Donnet, J. B., Wang, T. K., and Peng, J. C. M.: A comparison of diesel engine soot with carbon black, Carbon, 37, 1553-1565, https://doi.org/10.1016/S0008-6223(99)00035-4, 1999.

Harris, S. J., and Maricq, M. M.: Signature size distributions for diesel and gasoline engine exhaust particulate matter, J. Aerosol Sci., 32, 749-764, 2001.

Kalberer, M., Paulsen, D., Sax, M., Steinbacher, M., Dommen, J., Prévôt, A. S., Fisseha, R., Weingartner, E., Frankevich, V., and Zenobi, R.: Identification of polymers as major components of atmospheric organic aerosols, Science, 303, 1659-1662, 2004.

La Rocca, A., Bonatesta, F., Fay, M. W., and Campanella, F.: Characterisation of soot in oil from a gasoline direct injection engine using Transmission Electron Microscopy, Tribol. Int., 86, 77-84, https://doi.org/10.1016/j.triboint.2015.01.025, 2015.

Li, W. J., Zhang, D. Z., Shao, L. Y., Zhou, S. Z., and Wang, W. X.: Individual particle analysis of aerosols collected under haze and non-haze conditions at a high-elevation mountain site in the North China plain, Atmos. Chem. Phys., 11, 11733-11744, 10.5194/acp-11-11733-2011, 2011.

Philippin, S., Wiedensohler, A., and Stratmann, F.: Measurements of non-volatile fractions of pollution aerosols with an eight-tube volatility tandem differential mobility analyzer (VTDMA-8), J. Aerosol Sci., 35, 185-203, 10.1016/j.jaerosci.2003.07.004, 2004.

Villani, P., Picard, D., Marchand, N., and Laj, P.: Design and Validation of a 6-Volatility Tandem Differential Mobility Analyzer (VTDMA), Aerosol Sci. Tech., 41, 898-906, 10.1080/02786820701534593, 2007.

Wang, Y., Li, Z., Zhang, Y., Du, W., Zhang, F., Tan, H., Xu, H., Fan, T., Jin, X., Fan, X., Dong, Z., Wang, Q., and Sun, Y.: Characterization of aerosol hygroscopicity, mixing state, and CCN activity at a suburban site in the central North China Plain, Atmos. Chem. Phys., 18, 11739-11752, 10.5194/acp-18-11739-2018, 2018.

Wang, Z., Wu, Z., Yue, D., Shang, D., Guo, S., Sun, J., Ding, A., Wang, L., Jiang, J., Guo, H., Gao, J., Cheung, H. C., Morawska, L., Keywood, M., and Hu, M.: New particle formation in China: Current knowledge and further directions, Sci. Total Environ., 577, 258-266, https://doi.org/10.1016/j.scitotenv.2016.10.177, 2017.

Wehner, B., Berghof, M., Cheng, Y. F., Achtert, P., Birmili, W., Nowak, A., Wiedensohler, A., Garland, R. M., Pöschl, U., and Hu, M.: Mixing state of nonvolatile aerosol particle fractions and comparison with light absorption in the polluted Beijing region, Journal of Geophysical Research Atmospheres, 114, 85-86, 2009.

Zhang, Y., Liu, H., Lei, S., Xu, W., Tian, Y., Yao, W., Liu, X., Liao, Q., Li, J., Chen, C., Sun, Y., Fu, P., Xin, J., Cao, J., Pan, X., and Wang, Z.: Mixing state of refractory black carbon in fog and haze at rural sites in winter on the North China Plain, Atmos. Chem. Phys., 21, 17631-17648, 10.5194/acp-21-17631-2021, 2021.

---

## Author Response (AR3)

We thank the editor for providing helpful suggestions to improve the manuscript. Below we have included our responses in red. In the revision of this manuscript, we have highlighted those changes accordingly.

Editor comments are in blue; manuscript text in black.
1) Your new text in lines 75 – 79 seems out of place and misleading.
Modern gasoline direct injection (GDI) vehicles can emit plentiful ultrafine BC containing (soot) particles in the ambient (La Rocca et al., 2015; Hu et al., 2021).
The paragraph is about mixing state of soot particle. However, the information on size range of BC particles should come first, after the sentence in that starts in l. 58 (Black carbon is the most...). Please add not only the references by La Rocca 2015, and Hu et al., 2021 but also a reference that reports on BC in larger particles. For example, you could add briefly some information you wrote in the author response to the first editor comment.
RE: The sentence "Modern gasoline …" is deleted. The sentence "Soot particles are abundant in both nucleation and accumulation modes (Li et al., 2011; Levy et al., 2013; La Rocca et al., 2015; Hu et al., 2021; Zhang et al., 2021)." is added in line 63-65. The references of Levy et al. (2013) and Zhang et al. (2021) suggests that soot particles in accumulation mode are also abundant in the ambient.

The tiny soot particles embedded in other material (such as sulfate) play a significant role in particle growth (Li et al., 2011).
This sentence implies that the soot itself leads to particle growth. This is not reflected in the article by Li et al., 2011. They only say that particle growth occurs on particles that contain soot – among other components. Please reword it or remove sentence.
RE: This sentence is deleted.

Investigating the mixing state of BC-containing particles and their factors in different modes are needed.
This sentence seems out of place or redundant after the preceding paragraph where the role of mixing state is already discussed.
RE: This sentence is deleted.

2) l. 258: These tiny soot particles in nucleation mode are mainly from modern vehicle emissions (La Rocca et al., 2015; Hu et al., 2021)
The referee requested that *" Again, the authors need to demonstrate that the very volatile nucleation mode particles contain soot before discussing the high volatility of nucleation-mode soot particles"*
The simple addition of your statement that these particles are composed of soot is not sufficient to address the referee's concern. As pointed out by the referee, commonly it is assumed that soot particles are not (necessarily) highly volatile. How can you be sure that it was the case in your study? A brief discussion of this may deserve a separate paragraph as it is essential for the conclusions of your study.

RE: The fraction of VV-mode particles is high for the nucleation-mode (40-nm and 80-nm) particles shown in Fig. 3. The VV-mode (SF≈0.4) residual soot size of most 40-nm particles after heating at 300ºC was about 16 nm. And the VV-mode residual soot size of most 80-nm particles after heating at 300ºC was about 32 nm. Hu et al. (2021) found ultrafine soot particles in diameter below 23 nm take up majority of particle emissions from a China VI vehicle. La Rocca et al. (2015) reported that primary soot particles from the emission of light-duty EURO IV GDI engine presented an average diameter of 36 nm with a mode of 32 nm. Moreover, our measurement site (Xingtai) is located in the severe polluted area with kinds of anthropogenic sources including the high intensity of vehicle emissions. Therefore, it is excepted that these soot in nucleation mode are plentiful and mainly from vehicle emissions. Extremely low-volatile organics are another possible component in this size. However, these extremely low-volatile organics are mainly formed in the forest area (Ehn et al., 2014), which is not in line with where we were measuring. We added more discussion in the manuscript.